



# Saudi Rainfall (SaRa): Hourly 0.1° Gridded Rainfall (1979–Present) for Saudi Arabia via Machine Learning Fusion of Satellite and Model Data

Xuetong Wang[1], Raied S. Alharbi[2], Oscar M. Baez-Villanueva[3], Amy Green[4,5], Matthew F. McCabe[1], Yoshihide Wada[1], Albert I.J.M. Van Dijk[6], Muhammad A. Abid[7,8], and Hylke E. Beck[1,9]

[1]King Abdullah University of Science and Technology (KAUST), Thuwal, Saudi Arabia
[2]Department of Civil Engineering, College of Engineering, King Saud University, Riyadh, Saudi Arabia
[3]Hydro-Climate Extremes Lab (H-CEL), Ghent University, Ghent, Belgium
[4]School of Engineering, Newcastle University, Newcastle upon Tyne, UK
[5]Tyndall Centre for Climate Change Research, Newcastle University, Newcastle upon Tyne, UK
[6]Fenner School of Environment & Society, Australian National University, Canberra, ACT, Australia
[7]Atmospheric, Oceanic and Planetary Physics (AOPP), Department of Physics, University of Oxford, UK
[8]National Centre for Atmospheric Science (NCAS), UK
[9]GloH2O LLC, Princeton, USA

**Correspondence:** Hylke E. Beck (hylke.beck@gloh2o.org)

**Abstract.** We introduce Saudi Rainfall (SaRa), a gridded historical and near real-time precipitation ($P$) product specifically designed for the Arabian Peninsula, one of the most arid, water-stressed, and data-sparse regions on Earth. The product has an hourly 0.1° resolution spanning from 1979 to the present and is continuously updated with a latency of less than two hours. The algorithm underpinning the product involves 18 machine learning model stacks trained for different combinations of satellite and (re)analysis $P$ products along with several static predictors. As a training target, hourly and daily $P$ observations from gauges in Saudi Arabia ($n$=113) and globally ($n$=14,256) are used. To evaluate the performance of SaRa, we carried out the most comprehensive evaluation of gridded $P$ products in the region to date, using observations from independent gauges (excluded from training) in Saudi Arabia as a reference ($n$=119). Among the 20 evaluated $P$ products, our new product, SaRa, consistently ranked first across all evaluation metrics, including the Kling-Gupta Efficiency (KGE), correlation, bias, peak bias, wet days bias, and critical success index. Notably, SaRa achieved a median KGE — a summary statistic combining correlation, bias, and variability — of 0.36, while widely used non-gauge-based products such as CHIRP, ERA5, GSMaP V8, and IMERG-L V07 achieved values of $-0.07$, $0.21$, $-0.13$, and $-0.39$, respectively. SaRa also outperformed four gauge-based products such as CHIRPS V2, CPC Unified, IMERG-F V07, and MSWEP V2.8 which had median KGE values of $0.17$, $-0.03$, $0.29$, and $0.20$, respectively. Our new $P$ product — available at www.gloh2o.org/sara — addresses a crucial need in the Arabian Peninsula, providing a robust and reliable dataset to support hydrological modeling, water resource assessments, flood management, and climate research.





## 1 Introduction

The Kingdom of Saudi Arabia presents a striking hydrological paradox, experiencing periods of destructive flash floods and
acute water scarcity, often simultaneously. Flash floods, the Kingdom's most frequent natural hazard, occur on average seven
times a year across the country, incurring significant economic losses and social disruption(Al Saud, 2010). Particularly dev-
astating were the flash floods in Jeddah in 2009 and 2011, claiming 113 and 10 lives, respectively, and resulting in widespread
damage to property, averaging around 3 billion USD (Youssef et al., 2016). Moreover, climate change is projected to shift pre-
cipitation ($P$) patterns and increase atmospheric water vapor, potentially leading to more intense storms (Tabari and Willems,
2018; Almazroui et al., 2020; Fowler et al., 2021). Saudi Arabia also faces significant challenges in achieving water security
for its growing population. The arid climate of the region, combined with an increasing water demand driven by rapid urban-
ization, industrialization, and agricultural expansion, puts immense pressure on limited water resources (Al-Ibrahim, 1991;
Sultan et al., 2019). Effective management of these challenges requires accurate and timely $P$ estimates, as well as to assess
the impacts of climate change, develop adaptation and mitigation strategies, optimize water resources management, and im-
prove flash flood early warning systems. The development of such products is also crucial for achieving the objectives of the
National Water Strategy and Vision 2030 (Kingdom of Saudi Arabia, 2021), which aim to create a sustainable water sector
while providing cost-effective supply and high-quality services to foster economic and social development

Over the past few decades, a wide range of gridded $P$ products have been developed, each with unique design objectives,
spatial and temporal resolutions, coverage, latency, algorithms, and data sources, ranging from satellite to analysis, reanaly-
sis, gauges, and their combinations. Table 1 provides an overview of quasi- and fully-global products. In general, $P$ products
contain inherent errors and biases, making it important to assess their performance to determine their relative strengths, weak-
nesses, and suitability for different applications and utility for particular regions and geographies . While several global studies
have assessed the performance of many of these products, typically using gauge $P$ observations as reference (e.g., Beck et al.,
2017; Sun et al., 2018; Nguyen et al., 2018), they have often excluded Saudi Arabia due to the scarcity of local $P$ observa-
tions. To date, only a limited number of evaluations have specifically focused on the Arabian Peninsula (Kheimi and Gutub,
2015; Mahmoud et al., 2018; El Kenawy and McCabe, 2016; El Kenawy et al., 2019; Al-Falahi et al., 2020; Helmi and Abdel-
hamed, 2023; Alharbi et al., 2024; Jazem Ghanim et al., 2024). Two of these studies assessed individual satellite $P$ products —
IMERG (Mahmoud et al., 2018) and PDIR-Now (Alharbi et al., 2024) — leaving questions about the comparative performance
of these products unresolved. Four other studies evaluated multiple satellite $P$ products, including CMORPH, GSMaP, PER-
SIANN, SM2RAIN-ASCAT, and TMPA 3B42 (Kheimi and Gutub, 2015; El Kenawy et al., 2019; Helmi and Abdelhamed,
2023; Jazem Ghanim et al., 2024). Notably, El Kenawy et al. (2019). Helmi and Abdelhamed (2023), and Jazem Ghanim et al.
(2024) reported that the products generally performed poorly and highlighted the need for caution when using them. Al-Falahi
et al. (2020) evaluated some of the aforementioned satellite $P$ products as well as the reanalysis ERA5 but only focused on the
highland region of Yemen. However, many satellite products evaluated in these studies have been superseded by newer, signif-
icantly improved versions. Additionally, several promising products have not been evaluated yet, including SM2RAIN-GPM
(Massari et al., 2020), MSWEP V2.8 (Beck et al., 2019b), and JRA-3Q (Kosaka et al., 2024).





Several gridded $P$ products, such as CHIRPS V2 (Funk et al., 2015), GPCP (Huffman et al., 2023), MSWEP V2.8 (Beck et al., 2019b), and SM2RAIN-GPM (Massari et al., 2020; Table 1), leverage multiple $P$-related data sources to obtain improved $P$ estimates. These products employ statistical methods to minimize errors and biases inherent in individual sources, thereby enhancing $P$ estimation performance across various regions, seasons, and temporal scales. Although these products generally outperform single-source $P$ products (e.g., Beck et al., 2017; Prakash, 2019; Shen et al., 2020), machine learning (ML) approaches are increasingly recognized for their ability to efficiently fuse multiple data sources, while mitigating errors and biases. A wide variety of ML models, often trained with gauge observations, have been used for $P$ estimation, including classical models such as multivariate linear regression (MLR), artificial neural networks (ANNs), support vector machines (SVMs), and random forests (RF) along with modern deep learning models such as convolutional neural networks (CNNs) and long short-term memory (LSTM) networks and hybrid models (see reviews by Hussein et al., 2022; Dotse et al., 2024; Papacharalampous et al., 2023; Xu et al., 2024). However, most ML studies on $P$ estimation have limitations in that: (i) they generally focus on a small region or catchment, which limits the usefulness and generalizability of the findings; (ii) they often focus on a monthly (rather than daily or sub-daily) time scale, which may not meet the needs of all applications; (iii) they develop models for either near real-time or historical $P$ purposes, but not both; (iv) they use gauge observations as predictors, which precludes near real-time model application, given that gauge observations are generally not available in near real-time; (v) they remain largely theoretical, often failing to offer a corresponding, accessible $P$ dataset for users and follow-up studies. Additionally, and crucially, no study has yet investigated the potential of ML to specifically enhance $P$ estimates in the Arabian Peninsula region.

Here, we introduce Saudi Rainfall (SaRa), a new gridded near real-time $P$ product with an hourly 0.1° resolution designed to overcome the aforementioned limitations. The product covers the Arabian Peninsula from 1979 to the present with a latency of less than 2 hours. It was derived using ML models trained on a vast database of hourly and daily gauge $P$ observations from around the world. The ML models are tailored to various $P$ product combinations to ensure optimal performance for each period and location. In the following section, we describe the data and methods underlying the product. Subsequently, we (i) evaluate the performance of the ML models, constructed using different gridded $P$ product combinations as predictors, (ii) assess SaRa's performance relative to 19 global $P$ products, (iii) examine the spatial patterns in performance, (iv) discuss the challenges of estimating $P$ in arid regions, and (v) present trends in average and extreme $P$ for the Arabian Peninsula based on SaRa.

## 2 Data and Methods

### 2.1 Gridded precipitation and air temperature products

Global gridded $P$ products (more details in Table 1) were used for two purposes: (i) as predictors to generate our new $P$ product for the Arabian Peninsula (SaRa) and (ii) to evaluate the performance of SaRa relative to other $P$ products. Gridded air temperature ($T$) data were also used as predictors to account for seasonal differences in error characteristics among products. Throughout this paper, we refer to these global gridded predictors as "dynamic" due to their temporal variability — in





contrast to predictors that are invariant in time, which are referred to as "static" (see below). We restricted our selection to $P$ products with a daily or sub-daily temporal resolution. The $P$ products included in our study originate from diverse sources, encompassing satellite observations, ground-based gauges, reanalyses, analyses, and combinations thereof. The ML models are trained to optimally merge the $P$ products and mitigate errors and biases using gauge $P$ data. The products selected as predictors for developing SaRa were primarily non-gauge-corrected $P$ datasets to avoid biasing predictor importance, particularly

in cases where the same stations used for training might have been employed to correct the respective $P$ products. We included both microwave-based (IMERG-L V07 and GSMaP-MVK V8) and infrared-based (PERSIANN-CCS-CDR and PDIR-Now) satellite products as predictors. Among the predictors, the only gauge-corrected product is the infrared-based satellite-product PERSIANN-CCS-CDR, which has been corrected at the monthly scale using the Global Precipitation Climatology Project (GPCP) product (V2.3; monthly 2.5° resolution; Adler et al., 2018). PERSIANN-CCS-CDR was used prior to 2000, before the

more accurate microwave-based satellite-products IMERG and GSMaP became available. For consistency, $P$ and $T$ estimates from ERA5 and GDAS were resampled from 0.25° to 0.1° using nearest neighbor to develop SaRa, while PERSIANN-CCS-CDR and PDIR-Now were resampled from 0.04° to 0.1° using averaging.

## 2.2 Static predictors

We used six static predictors to develop our new $P$ product (Table 2). The term "static" indicates that these predictors are not

time-dependent. Among these predictors, two are climate-related (Aridity Index, AI; and mean annual $P$, Pmean), one pertains to topography (Effective Terrain Height, ETH), and three are linked to geographic location (latitude, longitude, and absolute latitude; Lat, Lon, and AbsLat, respectively). AI represents the ratio of mean annual $P$ to potential evaporation (PET). ETH quantifies the orographic influence on $P$ patterns by smoothing the topography (Daly et al., 2008). We excluded slope from our predictors as it was strongly correlated with ETH. Air temperature was omitted because it is already included as a dynamic

predictor. Each static predictor was resampled by averaging to match the resolution of SaRa (0.1°).

## 2.3 Precipitation observations

We used $P$ observations for two purposes: (i) to train the ML models underpinning our new $P$ dataset, SaRa, and (ii) to evaluate the performance of SaRa relative to other $P$ products (Fig. 1). Although the SaRa product was specifically developed for the Arabian Peninsula, due to the lack of hourly $P$ data in the Arabian Peninsula, we trained the ML models using $P$ observations

from across the globe. This approach assumes that valuable insights from other regions can help optimize the merging of $P$ products and reduce errors and biases in the Arabian Peninsula.Essentially, knowledge is transferred from data-rich (gauged) to data-poor (ungauged) regions, akin to regionalization techniques typically used in hydrology to tackle Predictions in Ungauged Basin (PUB) problems, which is considered a "grand challenge" in hydrology (Sivapalan et al., 2003; Hrachowitz et al., 2013).

For Europe and the conterminous US, our ground-based $P$ data sources were gridded $P$ datasets based on gauge and radar

data. Specifically, we used the EUropean RADar CLIMatology (EURADCLIM) dataset (hourly 2-km resolution; 2010–2022; Overeem et al., 2023) for Europe and the Stage-IV dataset (hourly 4-km resolution; 2002–present; Lin and Mitchell, 2005) for the conterminous US. To ensure the highest data quality, we extracted time series only at gauge locations from these



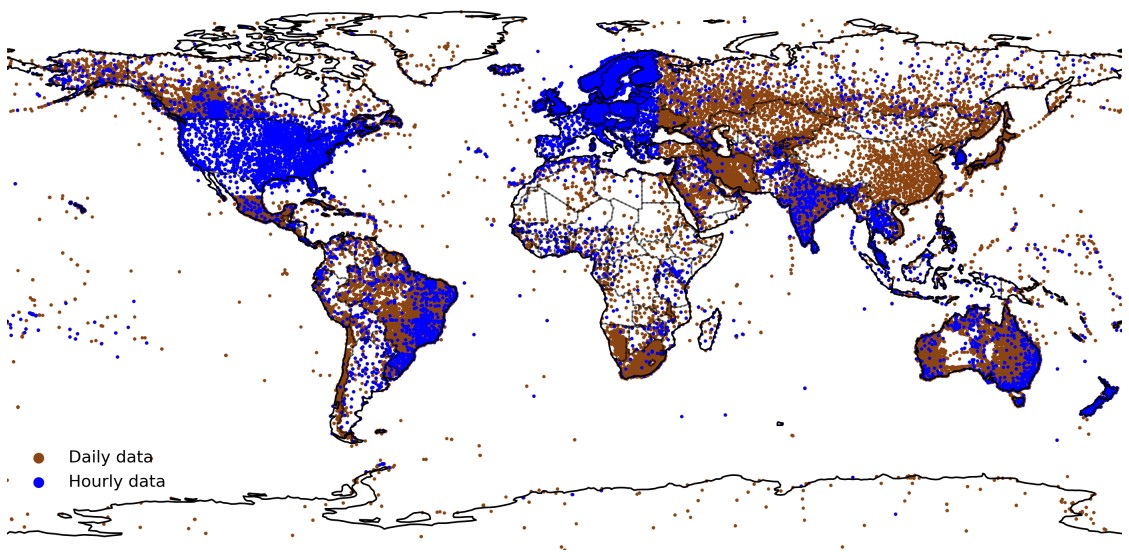

**Figure 1.** Map of the 104,346 gauges remaining after duplication checks and quality control. Stations with hourly data are shown in blue, while stations with daily data are shown in brown.

datasets after resampling the data to the resolution of SaRa ($0.1°$) using averaging. We opted for these gauge-radar datasets over direct gauge observations in these regions because they: (i) provide grid-cell averages with probability distributions (e.g.,
peak magnitudes and $P$ frequencies) matching those needed for SaRa; (ii) are expressed in UTC, avoiding temporal shifts; and (iii) have undergone extensive quality control.

Outside Europe and the conterminous US, we used daily and hourly gauge $P$ observations from various national, regional, and global data sources. The daily $P$ data sources include: (i) the Global Historical Climatology Network-Daily (GHCN-D) dataset (ftp.ncdc.noaa.gov/pub/data/ghcn/daily/; Menne et al., 2012; 40,867 gauges), (ii) the Global Summary Of the Day
(GSOD) dataset (https://data.noaa.gov; 9,904 gauges), (iii) the Latin American Climate Assessment & Dataset (LACA&D) dataset (http://lacad.ciifen.org/; 225 gauges), (iv) the Chile Climate Data Library (www.climatedatalibrary.cl; 712 gauges), and (v) national datasets for Brazil (10,963 gauges; www.snirh.gov.br/hidroweb/apresentacao), Mexico (3,908 gauges), Peru (255 gauges), Iran (3,100 gauges), and Saudi Arabia (459 gauges). The hourly $P$ observations encompassed 2,312 gauges from the Global Sub-Daily Rainfall (GSDR) dataset (Lewis et al., 2019) produced as part of the INTElligent use of climate models for
adaptatioN to non-Stationary hydrological Extremes (INTENSE) project (Blenkinsop et al., 2018), 12,585 from the Integrated Surface Database (ISD) stations (Smith et al., 2011), and national gauges from Brazil (289 gauges).

The training and evaluation of the ML model stacks used to generate SaRa was carried out for the period 2010–2024. We used this period instead of the full 1979 to the present period to reduce the significant memory requirements associated with hourly data. Additionally, EURADCLIM data starts in 2010, most Saudi Arabian gauge records begin in 2014, and GDAS data
starts in 2021.





## 2.4 Duplicates check and quality control

As we used $P$ observations from a diverse range of data sources, there was an increased risk of some gauges being included in multiple sources. To avoid over-representation of these gauges in the training set and ensure the same data were not used for both training and evaluation, we removed these duplicates. To this end, we iterated over all gauges, and if another gauge was located within a 2-km radius, we gave preference to the source we deemed most reliable. We used the following order of most to least reliable source: EURADCLIM, Stage-IV, GHCN-D, GSDR, ISD, Bolivia, Brazil, Chile, Mexico, Iran, LACA&D, GSOD, and MEWA.

$P$ observations are often subject to systematic, gross, and random errors (Kochendorfer et al., 2017; Tang et al., 2018), which can adversely affect the training and evaluation results. We identified and filtered out potentially erroneous gauges using the following five criteria: (i) non-zero minimum daily $P$; (ii) daily maximum less than 10 mm d$^{-1}$ or exceeding 1,825 mm d$^{-1}$ (the highest daily rainfall ever recorded; www.weather.gov/owp/hdsc_world_record); (iii) mean annual $P$ less than 5 mm yr$^{-1}$ or exceeding 10,000 mm yr$^{-1}$; (iv) fewer than five $P$ events (using a 1 mm d$^{-1}$ threshold); and/or (v) fewer than 365 daily values (not necessarily consecutive) during 2010–2024 (the training and evaluation period). In total, 75,833 gauges were discarded out of 104,346 after these steps.

## 2.5 SaRa precipitation estimation algorithm

The SaRa product was derived using different ML model stacks trained from different combinations of dynamic $P$ and $T$ predictors (Section 1) along with several static predictors (Section 2). Each model stack is comprised of four separate ML submodels (Fig. 2). The first submodel is a daily XGBoost model (Chen and Guestrin, 2016) trained using daily $P$ observations, leveraging the broad availability of $P$ observations globally and in the Arabian Peninsula (Fig. 1). The second submodel, also based on XGBoost, disaggregates the daily estimates to 3-hourly and is trained using 3-hourly $P$ observations, which are scarce in the Arabian Peninsul. As such, all of the 3-hourly disaggregation skill originates from other regions. As the resulting $P$ estimates tend to underestimate the variance (i.e., generate excessive drizzle and underestimate peaks) due to the regression towards the mean phenomenon (see, e.g., He et al., 2016; Ting, 2024), a third submodel based on Random Forest (RF; Breiman, 2001), corrects the 3-hourly $P$ probability distribution. The RF submodel is trained by separately sorting, for each gauge, (a) the 3-hourly estimates from the second submodel corrected using the daily estimates from the first submodel; and (b) the 3-hourly $P$ observations. To ensure the number of wet days and low $P$ intensities are also adequately corrected, the $P$ estimates are square-root transformed before being fed to the third submodel, and the output of the third submodel is squared. The fourth submodel disaggregates the 3-hourly estimates to hourly and also represents an XGBoost model, trained using hourly $P$ observations

The dynamic predictors span different time periods and different regions (see Table 1), so we cannot use a single ML model stack for every time step and grid-cell. We therefore trained a total of 18 different ML model stacks with various combinations of dynamic predictors (Table 3). These model stacks are used based on the available dynamic predictors for a specific time and location (Figure 3), with preference given to the model stack with the lowest number (e.g., model_01 is preferred over



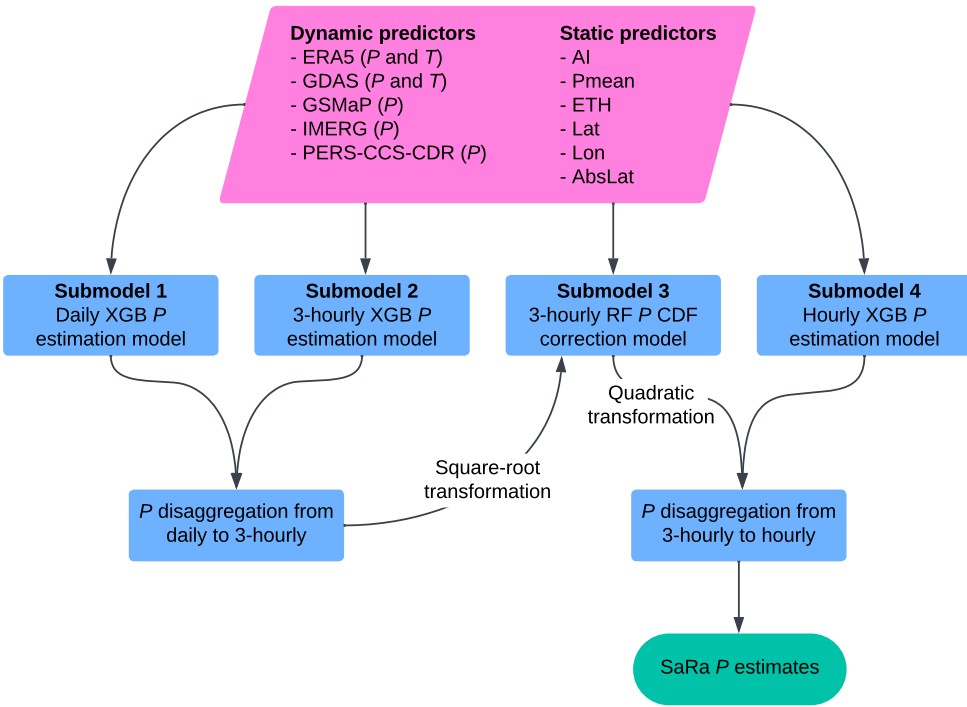

**Figure 2.** Flowchart of ML model stacks used to produce the new SaRa $P$ product presented in this study.

model_02). The final SaRa $P$ estimates were generated by iterating over all 0.1° grid-cells, loading all dynamic and static
predictors, and then applying the preferred ML model stack. To avoid temporal discontinuities, such as around 1983 when
PERSIANN-CCS-CDR was introduced or in 2000 when IMERG and GSMaP were introduced, the outputs from model_04 and
model_05 were harmonized with the outputs of model_01, which we consider the reference due to its long record (from 2000
to 5 days prior to the present) and high accuracy (owing to the availability of ERA5, IMERG, and GSMaP). The harmonization
process involved: (i) detrending the time series by dividing by the moving annual average, (ii) cumulative distribution function
(CDF) matching, and (iii) multiplying the result by the moving annual average (Figure 3). The detrending serves to avoid
amplification of trends in extreme $P$ (see, e.g., Cannon et al., 2015).

We implemented the RF models using the `scikit-learn` package and XGBoost models using the `XGBoost` package in
Python. The hyperparameters we used are summarized in Appendix A.

## 2.6   Training and evaluation

Both the training and evaluation were carried out for the period 2010–2024, aligning with the temporal coverage of EURAD-
CLIM and the Saudi gauge data. From the 28,513 gauges that passed the duplicates check and quality control (Section 2.4), we
allocated 50 % for training (14,256 gauges) and the remaining 50 % for evaluation. We trained submodel one using all available



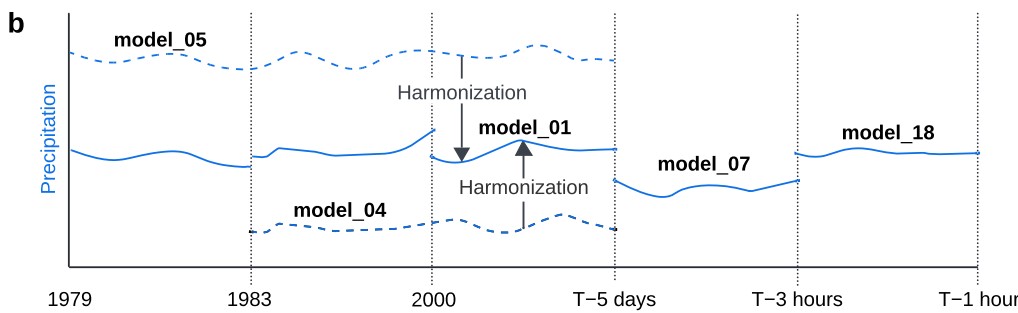

**Figure 3.** (a) Different ML model stacks were used for different periods and locations to account for differences in the spatio-temporal availability of the dynamic predictors. The primary ML model stack, the corresponding dynamic predictors, and the mean daily independent validation KGE (from Table 4) are also provided. Note that other ML model stacks may be used for a particular period when any dynamic predictor is not available. (b) Conceptual illustration of how $P$ estimates from different ML model stacks are combined, and how outputs from ML model stacks with long records are harmonized with the reference (model_01).

daily gauge data, while submodels two, three, and four were trained using gauges with hourly data (aggregated to 3-hourly for submodels two and three). The specific number of gauges used for training each model stack depends on the temporal span and spatial coverage of the dynamic predictors. Since GDAS covers such a short period (2021 to the present), any ML models incorporating GDAS were trained using a significantly smaller amount of observations.

The time stamps of the hourly $P$ observations may reflect the local time zone instead of Coordinated Universal Time (UTC), while the daily $P$ observations may represent accumulations that conclude at various times, not necessarily at midnight UTC (Yang et al., 2020). Such discrepancies can lead to temporal mismatches between the dynamic predictors on one side and the gauge $P$ observations on the other, thereby hindering satisfactory model training. To address this issue, we determined time shifts in the hourly and daily gauge $P$ data using the hourly satellite-based IMERG-L V07 and GSMaP-MVK V8 products, similar to Beck et al. (2019b). To determine hourly gauge data shifts, we shifted the gauge record by 1-hour increments from $-36$ to $+36$ and calculated the average Spearman correlations between shifted gauge data and IMERG-L V07/GSMaP-MVK V8



data. To determine daily gauge data shifts, we shifted the IMERG-L V07 and GSMaP-MVK v8 data separately by 1-hour
increments from $-36$ to $+36$, computed daily $P$ accumulations from the shifted IMERG-L V07 and GSMaP-MVK V8 data,
and calculated the average Spearman correlations between daily gauge record and shifted IMERG-L V07/GSMaP-MVK V8
data. The shifts that yielded the highest correlations were then used to recalculate daily values of the dynamic predictors for
training ML submodel one, as well as to shift the hourly gauge records for training submodels two and four.

The 119 evaluation gauges are completely independent and were not used to train the ML models, enabling a thorough
performance assessment of SaRa compared to other $P$ products (Table 4). We used several performance metrics for a compre-
hensive evaluation: (i) the Kling-Gupta Efficiency (KGE; Gupta et al., 2009; Kling et al., 2012), which is an aggregate metric
combining Pearson correlation ($r_{\text{dly}}$), overall bias ($\beta$), and variance bias ($\gamma$), (ii) monthly Pearson correlation ($r_{\text{mon}}$), (iii) peak
bias at the 99.5th percentile ($B_{\text{peak}}$; %), (iv) wet day bias ($B_{\text{wet days}}$, days; calculated using a $0.5$ mm d$^{-1}$ threshold), and
(v) Critical Success Index (CSI), measuring the ratio of hits to the sum of hits, false alarms, and misses for $P$ events exceeding
10 mm d$^{-1}$. These metrics, selected to encompass all important aspects of $P$ time series, were computed for each evaluation
gauge based on daily data, except for $r_{\text{mon}}$. We did not conduct an hourly evaluation due to the lack of hourly observations in
the Arabian Peninsula. For a detailed explanation of the performance metrics, including the equations, see Appendix B.

## 3  Results and Discussion

### 3.1  Performance of ML models

To generate the SaRa product, we trained 18 different ML model stacks with different $P$ product combinations (Table 3). The
performance of these models in terms of median KGE, calculated using daily $P$ data from independent evaluation gauges,
ranges from $0.03$ for model_05, which relies solely on one dynamic $P$ predictor (ERA5), to $0.43$ for model_06, which uses
four dynamic $P$ predictors (ERA5, GDAS, IMERG-L V07, and GSMaP-MVK V8; Table 4). These results align with our
expectation that models incorporating a larger number of dynamic predictors are able to extract complementary strengths
from them, thus exhibiting better performance. Model_01, based on three dynamic $P$ predictors (ERA5, IMERG-L V07, and
GSMaP-MVK V8), is arguably the most important ML model stack as it covers the largest portion of the record, from 2000
to 5 days prior to the present. Model_01 also performs well, achieving a median KGE of $0.36$, a median peak bias ($B_{\text{peak}}$) of
$-11.17$ %, a wet day bias ($B_{\text{wet days}}$) of $+1.42$ days, and a median weather event detection score (CSI$_{10\,\text{mm}}$) of $0.21$.

The $B_{\text{peak}}$ value of $-11.17$ % obtained by model_01 suggests a slight underestimation of high rainfall amounts (Table 4).
Although RF models are known to underestimate extremes due to the regression toward the mean phenomenon (see, e.g., He
et al., 2016), the most likely reason is that the gauge $P$ data used for evaluation represent point measurements, which typically
exhibit higher peaks than grid-cell averages (Ensor and Robeson, 2008). Thus, this apparent underestimation may primarily
reflect a scale discrepancy. Note that a significant portion of the training data comprises gridded gauge-radar $P$ data (see
Section 2.3), which have a $0.1°$ grid-cell scale consistent with SaRa. Similarly, the $B_{\text{wet days}}$ value of $+1.42$ days (Table 4),
indicating a minor overestimation of rainfall frequency, might also be attributable to this scale difference. Rainfall frequencies
are generally higher for grid-cell averages than for point measurements (Osborn and Hulme, 1997).





Among all the trained ML model stacks, model_05 performed the worst, with a median KGE of 0.05. Model_05 uses only one dynamic predictor (ERA5) and is designed primarily for the period before 1983, when only ERA5 data are available (Figure 3). It also performs worse than ERA5 alone (median KGE of 0.21), which is mainly attributable to poor bias ($\beta$) and
peak bias ($B_{\text{peak}}$) values. Fortunately, these issues are largely resolved during the harmonization step, where the outputs of model_04 and model_05 are harmonized with those of model_01, which is considered the reference (Figure 3).

## 3.2 Performance comparison with other gridded $P$ products

The primary model underpinning our newly developed SaRa product, model_01, exhibited superior performance across nearly all 12 performance metrics relative to all 19 other $P$ products (Table 4). Notably, model_01 attained a median KGE of 0.36,
significantly outperforming widely used $P$ products such as ERA5, JRA-3Q, CMORPH-RT, CHIRPS V2, IMERG-L V07, GSMaP MVK V8, and MSWEP V2.8, which obtained median KGE values of 0.21, 0.12, 0.21, 0.17, $-0.39$, $-0.13$, and 0.2, respectively. Additionally, model_01 demonstrated its ability to represent high $P$ intensities, exhibiting a low peak bias ($B_{\text{peak}}$) of $-11.17\%$, while the aforementioned other $P$ products showed higher biases of $-30.18\%$, $-18.02\%$, $-44.55\%$, $-27.82\%$, $+72.91\%$, $+44.7\%$, and $-28.65\%$. In terms of detecting $P$ events ($\text{CSI}_{10\,\text{mm}}$), model_01 scored a median value of
0.22, surpassing the other products with values ranging from 0.19 to 0.09.

Although SaRa was derived using an algorithm trained on gauge $P$ observations (from stations excluded in the evaluation), it was not directly corrected using gauge observations. Despite this, SaRa outperformed products entirely based on gauge observations (CPC Unified) or corrected using gauge observations (CHIRPS V2, IMERG-F V07, and MSWEP V2.8). This reflects the limited availability of gauge observations from Saudi Arabia in global databases like GHCN-D (Menne et al.,
2012; Kidd et al., 2017). Additionally, the lower performance of CHIRPS V2 and IMERG-F V07 may stem from their 5-day and monthly gauge corrections, respectively, which are less effective at improving performance on a daily time scale.

Among the purely (re)analysis-based products (ERA5, GDAS, and JRA-3Q), GDAS performed the best with a median KGE of 0.24, outperforming ERA5 (median KGE of 0.21) and JRA-3Q (median KGE of 0.12; Table 4). Among the purely microwave and infrared satellite-based products (CMORPH-RT and -RAW, GSMaP-MVK V8, IMERG-L and -E V07, and
SM2RAIN-GPM), CMORPH-RT emerged as the best (median KGE of 0.21), followed by SM2RAIN-GPM (median KGE of 0.18), and GSMaP-MVK V8 (median KGE of $-0.13$). Among the purely infrared satellite-based products (PERSIANN-CCS and PDIR-Now), PDIR-Now performed best. PDIR-Now also outperformed some microwave-based products (GSMaP-MVK V8 and IMERG-L and -E V07). Among the SM2RAIN products (SM2RAIN-GPM, SM2RAIN-ASCAT, and SM2RAIN-CCI), SM2RAIN-GPM obtained the best overall performance. In contrast, SM2RAIN-ASCAT and SM2RAIN-CCI exhibited
$\text{CSI}_{10\,\text{mm}}$ values of 0, underscoring the limited capability of algorithms that infer $P$ based on soil moisture signals to detect $P$ events $> 10$ mm d$^{-1}$ over the Arabian Peninsula. This is likely due to the extremely arid conditions in the Arabian Peninsula, where the soil dries out rapidly following $P$ events, reducing the effectiveness of soil moisture-based detection methods. These results are in agreement with Jazem Ghanim et al. (2024), who also found SM2RAIN-ASCAT to perform poorly in the Arabian Peninsula.



Previous studies that evaluated $P$ datasets for the Arabian Peninsula include Kheimi and Gutub (2015), Mahmoud et al. (2018), El Kenawy and McCabe (2016), El Kenawy et al. (2019), Al-Falahi et al. (2020), Helmi and Abdelhamed (2023), Alharbi et al. (2024), and Jazem Ghanim et al. (2024). Comparing our results to these studies is challenging because they assessed fewer $P$ products and used outdated versions. However, Alharbi et al. (2024) reported a mean $r_{\text{dly}}$ of 0.33 for PDIR-Now, which is comparable to our median value of 0.32 (Table 4). Similarly, Helmi and Abdelhamed (2023) reported KGE, $r_{\text{dly}}$, $r_{\text{mon}}$, and $\text{CSI}_{10\,\text{mm}}$ values for PERSIANN-CCS-CDR, CHIRPS, and IMERG-F that are also consistent with our results.

### 3.3 Spatial distribution of performance metrics

To illustrate SaRa's performance, Figure 4 shows the spatial distribution of its performance relative to other widely used $P$ datasets. Figures 4a and Figure 4b display the spatial distribution of SaRa's KGE and $\text{CSI}_{10,\text{mm}}$, respectively, as obtained by SaRa (model_01) during its evaluation over independent rain gauges across Saudi Arabia. The differences in performance between SaRa and three widely used products (CHIRPS V2, ERA5, and IMERG-L V07) are also shown for both metrics (Figures 4c–h), highlighting SaRa's overall superior performance. At first glance, the spatial distribution of both metrics appears random, with clusters of good performance adjacent clusters of poor performance, lacking a clear spatial organization. This randomness may be partly due to rain gauge measurement errors (Ciach, 2003; Daly et al., 2007; Sevruk et al., 2009), compounded by scale discrepancies between point-scale measurements and grid-scale averages (Yates et al., 2006).

To examine whether performance patterns are related to specific climatic or topographic factors, we calculated Spearman rank correlation coefficients between the climatic and topographic attributes of the evaluation rain gauges (Table 2) and the performance scores (Section 2.6) for SaRa's model_01, SaRa's model_06, MSWEP V2.8, IMERG-L V07, GSMaP MVK V8, and ERA5 (Table 5). Overall, the correlations were slightly weaker for model_01 and model_06 compared to the $P$ products, suggesting more stable performance, which is expected, given that the climatic and topographic variables were included as predictors in the models.

The metric $r_{\text{dly}}$, which evaluates the ability of models or products to estimate daily $P$ variability, is primarily sensitive to random errors and less influenced by systematic biases. In all models and products, $r_{\text{dly}}$ shows a positive correlation with aridity index (AI), indicating reduced performance in arid regions. This conforms with previous large-scale $P$ product evaluations (e.g., Beck et al., 2017; Sun et al., 2018; Abbas et al., 2025) and reflects the brief, intense, and localized nature of rainfall in such regions. Additionally, $r_{\text{dly}}$ exhibited negative correlations with effective terrain height (ETH) for all models and products, indicating lower performance in the mountainous southwest where orographic $P$ predominates. This is consistent with other $P$ product evaluations (e.g., Ebert et al., 2007; Derin et al., 2016; Beck et al., 2019a) and reflects the greater heterogeneity of $P$ in regions of complex terrain. Additionally, satellite retrieval of $P$ in mountainous regions is particularly challenging due to the shallow nature of orographic $P$ (Yamamoto et al., 2017; Adhikari and Behrangi, 2022).

Performance metrics related to systematic biases in magnitude ($\beta$ and $B_{\text{peak}}$) generally showed negative correlations with mean precipitation (Pmean) and positive correlations with the aridity index (AI). These correlations were particularly strong for IMERG-L V07, suggesting that this product could benefit from bias correction using climate indices.







**Figure 4.** Performance of the primary ML model underpinning SaRa (model_01) in terms of (a) Kling-Gupta Efficiency (KGE) and (b) detection of $P$ events ($CSI_{10\,mm}$). (c,e,g) difference in KGE between model_01 and CHIRPS V2, ERA5, and IMERG-L V07, respectively. (d,f,h) difference in $CSI_{10\,mm}$ between model_01 and CHIRPS V2, ERA5, and IMERG-L V07, respectively. Each data point represents an independent evaluation gauge ($n = 119$).





## 3.4 Challenges of precipitation estimation in arid regions

Although SaRa outperformed other $P$ products, its performance metrics might seem underwhelming. For instance, the daily
Pearson correlation ($r_{\mathrm{dly}}$) of 0.50 achieved by model_01 (SaRa's primary model; Table 4) indicates that only 25% ($100 \times 0.50^2$)
of the daily variability in $P$ observations is captured, while other products perform even worse. Similarly, a $\mathrm{CSI}_{10\,\mathrm{mm}}$ of 0.21
suggests a moderate ability to detect $P$ events exceeding 10 mm d$^{-1}$. These results align with prior large-scale evaluations
reporting lower accuracy of $P$ products in arid regions (e.g., Beck et al., 2017; Sun et al., 2018; Abbas et al., 2025), reflecting
the inherent challenge of precipitation estimation in these environments.

The challenges in arid regions stem from several key factors:

1. $P$ events in arid regions are typically short-duration, highly intense, and spatially localized, making them difficult to
   detect with satellites, simulate with models, or measure with gauges. This contrasts with temperate and cold climates,
   where $P$ events are generally longer-lasting, less intense, and spatially broader (El Kenawy et al., 2019; Ebert et al.,
   2007).

2. Reanalyses like ERA5 and JRA-55 and analyses like GDAS are based on numerical weather prediction (NWP) models,
   which struggle to simulate the complex convective processes that predominate in arid-region $P$, including deep convec-
   tion initiation, rapid cell dissipation, and the effects of dry boundary layers (Yano et al., 2018; Peters et al., 2019; Lin
   et al., 2022).

3. Virga $P$, which evaporates before reaching the ground, adds another layer of complexity. It is estimated to account for
nearly half of all events in some arid regions, leading to significant false detections by satellite radiometers (Wang et al.,
   2018).

4. Errors in gauge measurements — due to wind deflection, evaporation within the gauge, splashing, and wetting losses —
   can also play a significant role (Ciach, 2003; Daly et al., 2007; Sevruk et al., 2009). In arid regions, higher evaporation
   rates exacerbate wetting losses, and the sparse, short-lived nature of rainfall events likely amplifies sampling errors
(Villarini et al., 2008).

5. Discrepancies between point measurements from gauges and grid-cell averages derived from satellite or model products
   also contribute to lower performance scores (Yates et al., 2006; Ensor and Robeson, 2008). In arid regions, this mismatch
   is likely particularly significant, due to the highly localized nature of $P$.

6. Time shifts between daily $P$ totals from gauges and satellite or (re)analysis products further reduce performance scores
(Yang et al., 2020; Beck et al., 2019b). The boundary between daily totals from satellite or (re)analysis products is
   midnight UTC, whereas for daily gauge totals the time of the boundary varies depending on regional reporting practices.
   In Saudi Arabia, the boundary time was determined to be, on average, 05:00 AM UTC (08:00 AM local time; see
   Section 2.6). Consequently, there is a $100 \times 5/24 = 21$ % chance that an hourly event will be assigned to the "wrong"
   day.



7. In arid regions, where rainfall is infrequent and measurements are often considered unnecessary, the number of stations is usually limited (Menne et al., 2012; Kidd et al., 2017). As a result, $P$ products may not be evaluated in these areas during development, potentially leading to lower performance.

However, it should be kept in mind that a hypothetical baseline $P$ product predicting only the mean would achieve a KGE of $-0.41$ (Knoben et al., 2019), making SaRa's model_01 median KGE of 0.36 quite reasonable, situated between this baseline and an (unattainable) perfect score (KGE of 1). Furthermore, performance improves markedly when data is averaged over longer periods, as evidenced by the median monthly Pearson correlation ($r_{mon}$) of 0.71 for model_01 (Table 4), indicating that more than twice as much variability is captured at the monthly scale compared to the daily scale. Similarly, performance is enhanced at larger spatial scales, for example when computing regional averages or when driving a hydrological model for a catchment. This improved performance reflects the reduced impact of random and gross errors due to spatial aggregations within catchments and regions (O and Foelsche, 2019).

## 3.5 Precipitation climatology and trends in Saudi Arabia

According to the newly developed SaRa $P$ product, the mean annual $P$ for Saudi Arabia during the period 1991–2020 is 54 mm yr$^{-1}$ (Fig. 5a). However, although SaRa's median $\beta$ score is 1.03 (Table 4), indicating negligible bias according to this metric, the gauge-based mean $P$ is nonetheless 18 % higher than the SaRa-based mean $P$ across all evaluation gauges, likely reflecting the tendency of ML models to attenuate extremes (see, e.g., He et al., 2016; Ting, 2024). Consequently, an adjusted estimate of $54 \times 1.18 = 64$ mm yr$^{-1}$ may represent the best estimate for mean annual $P$ in Saudi Arabia. This value is significantly lower than the estimate of 102 mm yr$^{-1}$ for the period 1991–2020 from the Climatic Research Unit (CRU) gridded Time Series (TS) dataset (Harris et al., 2020), as published on the World Bank website (https://climateknowledgeportal.worldbank.org/country/saudi-arabia/climate-data-historical). However, the CRU climatology is based on interpolation of a small number of gauges (approximately ten) in Saudi Arabia (New et al., 1999), whereas SaRa was trained using 113 stations in the country. The mean annual $P$ estimate of 84 mm yr$^{-1}$ from Almazroui (2011), based on TMPA 3B42 (Huffman et al., 2007) bias-corrected using 29 stations, comes closer to our estimate, although it covers a different period (1998–2009).

In addition to annual totals, SaRa provides insights into $P$ frequency and peak $P$ magnitudes. The average annual maximum daily $P$ in Saudi Arabia is 19 mm d$^{-1}$ (Fig. 5c). While this value is modest compared to the global best-estimate mean of 56 mm d$^{-1}$ (from the global gauge-based Precipitation Probability DISTribution — PPDIST — dataset V1.0; Beck et al., 2020), such $P$ extremes can still cause severe flooding due to the region's low soil infiltration capacity, sparse vegetation, and insufficient flood management infrastructure (Othman et al., 2023). The average annual maximum hourly $P$ is 6.9 mm h$^{-1}$. On average, Saudi Arabia experiences 10 rainy days per year (defined as days with $P \geq 0.5$ mm d$^{-1}$; Fig. 5e) and 51 rainy hours per year (defined as hours with $P \geq 0.1$ mm h$^{-1}$). For context, the global average number of rainy days per year is 30, based on PPDIST, using the same threshold of 0.5 mm d$^{-1}$. Across all metrics — mean annual $P$, $P$ frequency, and $P$ extremes — the highest values occur along the western slopes of the Asir Mountains, where orographic effects enhance $P$ (Hasanean and Almazroui, 2015).





Trend analysis from 1979 to 2023 based on SaRa reveal declines in mean annual $P$, daily $P$ frequency, and annual maximum daily $P$ at rates of $-0.50$, $-0.11$, and $-0.58$ % yr$^{-1}$, respectively (see Figs. 5b, 5d, and 5f, respectively). Over the 45-year period, these rates correspond to cumulative reductions of $-22.5$ %, $-5.0$ %, and $-26.1$ %, respectively. Our results align with Almazroui (2020), who reported a mean annual $P$ trend of $-0.65$ % yr$^{-1}$ during 1978–2019 based on 25 stations in Saudi Arabia. They are also consistent with Munir et al. (2025), who analyzed Standardized Precipitation Index (SPI) time series from 28 stations in Saudi Arabia for 1985–2023, finding negative trends at 16 stations and positive trends at 10. Furthermore, the strong declines observed in southeastern Saudi Arabia (Fig. 5b) align with Patlakas et al. (2021), who analyzed trends for 1986–2015 using a regional atmospheric model. However, these trend estimates are subject to significant uncertainty due to considerable interannual variability, as well as substantial uncertainties in gauge, model, and satellite-based $P$ estimates (see Section 3.4). While SaRa outperforms other products (Table 4), it also remains subject to considerable uncertainty. These trends result from multiple factors, including internal climate variability, external natural influences, and human-induced climate change. Interestingly, future projections from climate models in the sixth phase of the Coupled Model Intercomparison Project (CMIP6) suggest that increases in all three metrics are likely across most regions in Saudi Arabia (Iturbide et al., 2021; Intergovernmental Panel on Climate Change (IPCC), 2023).

## 4 Conclusions

The SaRa dataset, a high-resolution, gridded and near real-time $P$ product was developed to satisfy the critical need for more accurate and robust $P$ data in the Arabian Peninsula. SaRa offers hourly data at a 0.1° resolution spanning from 1979 to the present, with a latency of less than two hours. The algorithm underpinning the product involves 18 ML model stacks tailored to various $P$ product combinations to ensure optimal performance for each period and location. These models were trained using daily and hourly gauge $P$ observations from across the globe to enhance their accuracy. Our primary findings are summarized as follows:

1. Among the 18 model stacks, model_06, which incorporates four dynamic predictors (ERA5, GDAS, IMERG-L V07, GSMaP-MVK V8), achieved the highest median KGE (0.43). model_01, the primary model spanning the longest period (2000 to five days before the present) and incorporating three dynamic predictors (ERA5, IMERG-L V07, GSMaP-MVK V8), also demonstrated strong performance with a median KGE of 0.36. Most models showed minimal biases in peak rainfall and wet day frequency, reinforcing the idea that product performance can be enhanced by combining complementary strengths of diverse $P$ datasets. However, model_05, relying solely on ERA5, performed relatively poorly (median KGE of 0.03) due to significant bias issues, although these were mitigated through harmonization with model_01.

2. We carried out the most comprehensive daily evaluation of gridded $P$ products in the Arabian Peninsula to date. SaRa outperformed all 19 other gridded $P$ products across nearly all 12 performance metrics, achieving a median KGE of 0.36. Notably, it demonstrated superior event detection and lower peak bias compared to state-of-the-art products such



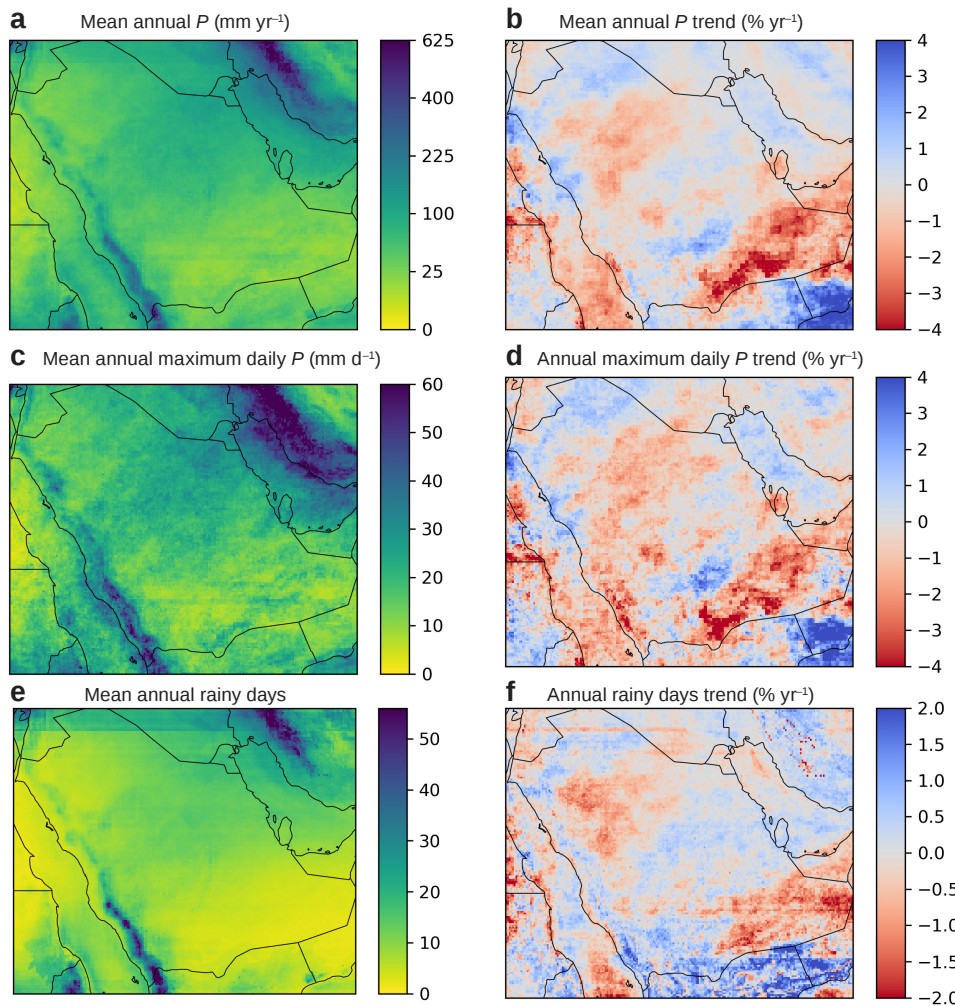

**Figure 5.** Mean and trend values (1979–2023) based on the new SaRa $P$ product for (a–b) mean annual $P$, (c–d) annual maximum daily $P$, and (e–f) annual number of wet days (using a threshold of 0.5 mm d$^{-1}$). Note that panel (a) uses a non-linear color scale.



as ERA5, JRA-3Q, CMORPH, and IMERG-F V07. Among the evaluated products, microwave-based satellite products
       generally performed better than infrared-based satellite products and (re)analyses.

3.  The spatial performance analysis of SaRa's model_01 across Saudi Arabia, based on KGE and $CSI_{10,mm}$ metrics, reveals a
    seemingly random distribution of performance, with clusters of high and low performance influenced by rain gauge errors
    and scale discrepancies between point and grid measurements. Correlations with climatic and topographic variables

suggest relatively stable performance, as these were incorporated as predictors in the model. Performance in estimating
       daily $P$ variability ($r_{dly}$) decreases in arid regions and mountainous areas, reflecting challenges with localized, intense
       rainfall and shallow orographic $P$.

4.  Despite outperforming all other products, SaRa's performance may nonetheless seem somewhat underwhelming. These
    results highlight the inherent difficulty in estimating $P$ in arid regions, where events are typically localized, brief, and

intense. For example, SaRa's primary model, model_01, captured only 25 % of daily $P$ variability, and the other $P$
       products less, possibly due to challenges including virga, short-duration and highly variable rainfall. However, it is
       worth noting that achieving perfect scores is impossible due to inherent gauge errors, scale discrepancies, and time shifts
       in daily accumulations.

5.  Mean annual $P$ across Saudi Arabia was estimated as 64 mm year$^{-1}$ over the period from 1991–2020, which is signif-

icantly lower than prior estimates based solely on rain gauges. Saudi Arabia averages 10 rainy days and 51 rainy hours
       annually, with higher and more frequent $P$ in the southwestern Asir mountains due to orographic effects. From 1979 to
       2023, $P$ trends show a decline in annual totals, frequency, and extremes (up to $-26.1$ %), driven by climate variability
       and anthropogenic factors. Climate projections suggest potential future increases, highlighting the need and value of
       data-driven $P$ approaches in resolving potential discrepancies in $P$ distributions and spatio-temporal patterns.

Our study addresses the long-standing need for more accurate $P$ estimates and provides a comprehensive evaluation of
gridded $P$ products in Saudi Arabia, one of the most arid, water-stressed, and data-sparse regions on Earth. The SaRa dataset,
available for use and distribution at www.gloh2o.org/sara, equips researchers, professionals, and policymakers with the tools
needed to tackle pressing environmental and socio-economic challenges in Saudi Arabia, and serves as a potential framework
for filling this data gap in other arid and dryland regions. The product delivers a high-resolution, near real-time resource
designed to support a diverse range of applications, including water resource management, hydrological modeling, agricultural
planning, disaster risk reduction, and climate studies.

*Code availability.* The code used to generate the results of this study is available from the corresponding author upon request.

*Data availability.* CPC Unified is available on the NOAA NOAA Physical Sciences Laboratory (PSL) website (https://psl.noaa.gov/data/
gridded/data.cpc.globalprecip.html). IMERG V07 can be accessed from the NASA Global Precipitation Measurement (GPM) website



(https://gpm.nasa.gov/data). JRA-3Q is available via the National Center for Atmospheric Research (NCAR) Research Data Archive (RDA; https://rda.ucar.edu/datasets/ds640000/dataaccess). SM2RAIN-ASCAT, SM2RAIN-CCI, and GPM+SM2RAIN are hosted on Zenodo (https://zenodo.org/records/10376109, https://zenodo.org/records/1305021, and https://zenodo.org/records/3854817, respectively). ERA5 data can be obtained from the Copernicus Climate Data Store (CDS; https://cds.climate.copernicus.eu/datasets/reanalysis-era5-single-levels?tab=overview). CHIRP and CHIRPS V2 are available via the University of California, Santa Barbara, Climate Hazards Center (CHC) web-

site (https://www.chc.ucsb.edu/data/chirps/). MSWEP V2.8 can be accessed via the GloH2O website (https://www.gloh2o.org/mswep/). PERSIANN-CCS-CDR and PDIR-Now are accessible via the Center for Hydrometeorology and Remote Sensing (CHRS) website (https://chrsdata.eng.uci.edu/). CHELSA can be accessed via https://chelsa-climate.org.

## Appendix A: ML model hyperparameters

The hyperparameters used for the RF and XGBoost models are described in Tables A2 and A1, respectively. These hyperpa-

rameters were selected to balance model complexity and training time, while also minimizing the risk of overfitting.

## Appendix B: Performance metrics calculation

The Kling-Gupta Efficiency (KGE) is given by:

$$\text{KGE} = 1 - \sqrt{(r-1)^2 + (\beta-1)^2 + (\gamma-1)^2}, \tag{B1}$$

where $r$ is the Pearson correlation coefficient, $\beta$ is the overall bias (mean simulated value to mean observed value), and $\gamma$ is

the variance bias (ratio of simulated variance to observed variance).

The Critical Success Index (CSI) is calculated for events exceeding a threshold of 10 mm d$^{-1}$ as:

$$\text{CSI} = \frac{H}{H + M + F + 10^{-9}}, \tag{B2}$$

where $H$ is the number of hits (correctly predicted events), $M$ is the number of misses (events missed by the product), and $F$ is the number of false alarms (incorrectly predicted events). An epsilon value is added to prevent division by zero.

Peak bias at the 99.5th percentile ($B_{\text{peak}}$; %) is calculated as the percentage difference between the 99.5th percentile of the estimated and observed data:

$$B_{\text{peak}} = 100 \times \frac{P_{99.5} - O_{99.5}}{O_{99.5}}, \tag{B3}$$

where $P_{99.5}$ and $O_{99.5}$ are the 99.5th percentiles of the estimated and observed values, respectively.

Wet day bias ($B_{\text{wet days}}$; days) is calculated as the percentage difference in the number of wet days (days exceeding a

0.5 mm d$^{-1}$ threshold) between simulated and observed data:

$$B_{\text{wet days}} = 365.25 \times \frac{P - O}{N}, \tag{B4}$$

where $P$ and $O$ are the number of wet days in the estimated and observed time series, respectively, and $N$ is the total number of values.



*Author contributions.* Author contributions. XW: modeling, analysis, visualization, and writing. HEB: initial idea, conceptualization, mod-
eling, analysis, writing, and project management. All coauthors contributed to writing, revising, and refining the manuscript.

*Competing interests.* The authors declare that they have no conflict of interest.

*Acknowledgements.* We sincerely acknowledge the developers of the datasets listed in Tables 1 and 2 for their efforts in creating and sharing
these valuable resources. We gratefully acknowledge Prof. Mansour Almazroui (King Abdulaziz University, Jeddah, Saudi Arabia) for sug-
gesting the name 'SaRa' for the product. For part of the analysis, we utilized the Shaheen III supercomputer, managed by the Supercomputing
Core Laboratory at King Abdullah University of Science and Technology (KAUST) in Thuwal, Saudi Arabia. This research was supported
in part by funding from KAUST's Center of Excellence for Generative AI, under award number 5940. Additionally, part of the research was
supported by the KAUST/MEWA Strategic Partnership Agreement (SPA) for Water, under award numbers 6110 and 6111.



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

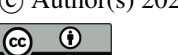



**Table 1.** Overview of quasi- and fully-global $P$ products used in this study. Abbreviations: P=precipitation; T=temperature; S=satellite; G=gauge; Re=reanalysis; A=analysis; NRT=near-real-time; Pred=used as predictor to generate SaRa; and Eval=included in the performance assessment. The column for spatial coverage denotes "Global" for complete global coverage including ocean regions, and "Land" for coverage limited to terrestrial areas. Version information unavailable for most products.

| Data | Version | Variables | Data Source | Resolution | | Coverage | | Time Latency | References/URL | Utilization |
|---|---|---|---|---|---|---|---|---|---|---|
| | | | | Temporal | Spatial | Temporal | Spatial | | | |
| IMERG-E | V07 | $P$ | S | 30 min | 0.1° | 2000–NRT | 60° N/S | ~4 hours | Huffman et al. (2019) | Eval, Pred |
| IMERG-L | V07 | $P$ | S | 30 min | 0.1° | 2000–NRT | 60° N/S | ~12 hours | Huffman et al. (2019) | Eval, Pred |
| IMERG-F | V07 | $P$ | S, G | 30 min | 0.1° | 2000–NRT | 60° N/S | ~3 months | Huffman et al. (2019) | Eval |
| GSMaP-NRT | V8 | $P$ | S | hourly | 0.1° | 2000–NRT | 60° N/S | ~4 hours | Kubota et al. (2020) | Eval, Pred |
| GSMaP-MVK | V8 | $P$ | S | hourly | 0.1° | 2000–NRT | 60° N/S | ~3 days | Kubota et al. (2020) | Eval, Pred |
| ERA5 | / | $P$ | Re | hourly | 0.25° | 1940–NRT | Global | ~5 days | Hersbach et al. (2020) | Eval, Pred |
| ERA5 | / | $T$ | Re | hourly | 0.25° | 1940–NRT | Global | ~5 days | Hersbach et al. (2020), | Pred |
| GDAS | / | $P$ | A | hourly | 0.25° | 2001–NRT | Global | ~3–6 hours | [1] | Eval, Pred |
| GDAS | / | $T$ | A | hourly | 0.25° | 2001–NRT | Global | ~3–6 hours | [1] | Pred |
| PDIR-Now | / | $P$ | S | hourly | 0.04° | 2000–NRT | 60° N/S | ~100 minutes | Nguyen et al. (2020) | Eval, Pred |
| PERSIANN-CCS-CDR | / | $P$ | S, G | 3-hourly | 0.04° | 1983–2021 | 60° N/S | / | Sadeghi et al. (2021) | Eval, Pred |
| JRA-3Q | / | $P$ | Re | 3-hourly | ~40 km | 1947–NRT | Global | ~20 days | Kosaka et al. (2024) | Eval |
| CMORPH-RAW | / | $P$ | S | 30 min | ~8 km | 2019–NRT | 60°N/S | ~4 hours | Joyce et al. (2004) | Eval |
| CMORPH-RT | / | $P$ | S | 30 min | ~8 km | 2019–NRT | 60° N/S | ~4 hours | Xie et al. (2017) | Eval |
| PERSIANN-CCS | / | $P$ | S | Hourly | 0.04° | 2003–NRT | 60° N/S | ~90 minutes | Hong et al. (2004) | Eval |
| CPC Unified | / | $P$ | G | Daily | 0.5° | 1979–NRT | Land | ~1 day | Chen et al. (2008) | Eval |
| SM2RAIN-CCI | / | $P$ | S | Daily | 0.25° | 1998–2015 | Land | / | Ciabatta et al. (2018) | Eval |
| SM2RAIN-ASCAT | / | $P$ | S | Daily | 0.1° | 2007–2021 | Land | / | Brocca et al. (2019) | Eval |
| SM2RAIN-GPM | / | $P$ | S | Daily | 0.25° | 2007–2018 | Land | / | Massari et al. (2020) | Eval |
| CHIRP | V2 | $P$ | S, Re, A | Daily | 0.05° | 1981–NRT | Land, 50° N/S | ~6 days | Funk et al. (2015) | Eval |
| CHIRPS | V2 | $P$ | S, G, Re, A | Daily | 0.05° | 1981–NRT | Land, 50° N/S | 2 weeks | Funk et al. (2015) | Eval |
| MSWEP | V2.8 | $P$ | S, G, Re, A | 3-hourly | 0.1° | 1979–NRT | Global | ~3 hours | Beck et al. (2019b) | Eval |
| SaRa | V1 | $P$ | S, G, Re, A | Hourly | 0.1° | 1979–NRT | Global | ~100 minutes | This paper | Eval |

[1] https://www.ncei.noaa.gov/products/weather-climate-models/global-data-assimilation



**Table 2.** Overview of the static predictors used in the ML models underpinning our new $P$ product, SaRa.

| Name (units) | Data Source(s) | Description |
| --- | --- | --- |
| AI ($-$) | Mean annual $P$ from CHELSA V2.1 (1-km resolution; Karger et al., 2017) and PET from Trabucco and Zomer (2018, 1 km) for land and ERA5 ($0.25°$) for ocean | Aridity index (AI) calculated as ratio of $P$ to PET |
| Pmean ($-$) | CHELSA V2.1 (1-km resolution; Karger et al., 2017) | Mean annual $P$ |
| ETH (m) | Global Multi-resolution Terrain Elevation Data (GMTED) 2010 (Danielson and Gesch, 2011) | Effective Terrain Height (ETH) calculated following Daly et al. (2008) |
| Lat ($°$) | / | Latitude |
| Lon ($°$) | / | Longitude |
| AbsLat ($°$) | / | Absolute latitude |

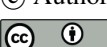



**Table 3.** The dynamic predictors incorporated in the different ML model stacks.

| | ERA5 $P$ | ERA5 $T$ | GDAS $P$ | GDAS $T$ | IMERG-L V07 | GSMaP-MVK V8 | P-CCS-CDR | PDIR-Now |
|---|---|---|---|---|---|---|---|---|
| 01 | ✓ | ✓ | ✗ | ✗ | ✓ | ✓ | ✗ | ✗ |
| 02 | ✓ | ✓ | ✗ | ✗ | ✓ | ✗ | ✗ | ✗ |
| 03 | ✓ | ✓ | ✗ | ✗ | ✗ | ✓ | ✗ | ✗ |
| 04 | ✓ | ✓ | ✗ | ✗ | ✗ | ✗ | ✓ | ✗ |
| 05 | ✓ | ✓ | ✓ | ✗ | ✗ | ✓ | ✗ | ✗ |
| 06 | ✓ | ✗ | ✓ | ✓ | ✓ | ✓ | ✗ | ✗ |
| 07 | ✗ | ✓ | ✓ | ✓ | ✓ | ✗ | ✗ | ✗ |
| 08 | ✓ | ✓ | ✓ | ✓ | ✓ | ✓ | ✗ | ✗ |
| 09 | ✓ | ✗ | ✓ | ✓ | ✗ | ✗ | ✗ | ✗ |
| 10 | ✗ | ✗ | ✗ | ✓ | ✓ | ✓ | ✗ | ✗ |
| 11 | ✗ | ✗ | ✗ | ✗ | ✗ | ✓ | ✗ | ✗ |
| 12 | ✗ | ✗ | ✗ | ✗ | ✓ | ✗ | ✗ | ✗ |
| 13 | ✗ | ✗ | ✗ | ✗ | ✓ | ✓ | ✗ | ✗ |
| 14 | ✗ | ✗ | ✓ | ✓ | ✗ | ✓ | ✗ | ✗ |
| 15 | ✗ | ✓ | ✓ | ✓ | ✗ | ✗ | ✗ | ✓ |
| 16 | ✓ | ✗ | ✓ | ✓ | ✗ | ✗ | ✗ | ✗ |
| 17 | ✗ | ✗ | ✗ | ✗ | ✗ | ✗ | ✗ | ✗ |
| 18 | ✗ | ✗ | ✗ | ✗ | ✗ | ✗ | ✗ | ✓ |





**Table 4.** The performance of the ML model stacks underlying our new $P$ product SaRa (models_01–18) and other state-of-the-art $P$ products sorted in descending order of median KGE. The values represent medians calculated over all independent evaluation gauges in Saudi Arabia. Note that since the different $P$ products span different temporal periods, the specific evaluation data used for the evaluation differs between $P$ products. The unit for $B_{peak}$ is %, and the unit for $B_{wet\,days}$ is the number of days. $N_{obs}$ indicates the number of stations that were used to assess each product.

| | KGE | $r_{dly}$ | $\beta$ | $|\beta-1|$ | $\gamma$ | $|\gamma-1|$ | $r_{mon}$ | $B_{peak}$ | $|B_{peak}|$ | $B_{wet\,days}$ | $|B_{wet\,days}|$ | $CSI_{10\,mm}$ | $N_{obs}$ |
|---|---|---|---|---|---|---|---|---|---|---|---|---|---|
| model_06 | 0.43 | 0.55 | 1.03 | 0.22 | 0.94 | 0.15 | 0.75 | 0.83 | 24.77 | 0.00 | 2.89 | 0.22 | 119 |
| model_08 | 0.41 | 0.53 | 0.94 | 0.27 | 0.95 | 0.13 | 0.75 | -5.68 | 28.49 | 0.39 | 2.48 | 0.20 | 119 |
| model_03 | 0.39 | 0.49 | 0.94 | 0.24 | 0.98 | 0.13 | 0.69 | -16.56 | 26.77 | 1.38 | 2.57 | 0.20 | 119 |
| model_07 | 0.39 | 0.51 | 1.01 | 0.25 | 0.94 | 0.14 | 0.73 | 3.64 | 27.67 | 1.06 | 3.21 | 0.21 | 119 |
| model_09 | 0.39 | 0.51 | 0.88 | 0.28 | 0.99 | 0.15 | 0.74 | -14.60 | 30.84 | -0.39 | 3.17 | 0.22 | 119 |
| model_11 | 0.37 | 0.53 | 1.02 | 0.28 | 0.93 | 0.15 | 0.73 | -3.34 | 28.28 | 0.00 | 3.12 | 0.22 | 119 |
| model_01 | 0.36 | 0.50 | 1.03 | 0.24 | 0.96 | 0.12 | 0.71 | -11.17 | 27.01 | 1.42 | 2.58 | 0.21 | 119 |
| model_02 | 0.36 | 0.46 | 0.93 | 0.27 | 1.02 | 0.12 | 0.69 | -19.14 | 31.26 | 0.43 | 2.08 | 0.20 | 119 |
| model_10 | 0.34 | 0.48 | 1.02 | 0.28 | 0.96 | 0.13 | 0.72 | -10.42 | 33.33 | 0.34 | 2.62 | 0.19 | 119 |
| model_12 | 0.32 | 0.46 | 1.01 | 0.25 | 0.89 | 0.13 | 0.64 | -7.59 | 26.20 | 3.44 | 4.35 | 0.18 | 119 |
| model_14 | 0.32 | 0.44 | 1.06 | 0.21 | 0.84 | 0.18 | 0.61 | -10.88 | 22.57 | 3.71 | 4.31 | 0.17 | 119 |
| model_15 | 0.31 | 0.45 | 0.93 | 0.30 | 0.90 | 0.17 | 0.72 | -15.25 | 30.62 | 1.68 | 3.28 | 0.16 | 119 |
| model_13 | 0.29 | 0.41 | 1.04 | 0.25 | 0.87 | 0.15 | 0.57 | -3.85 | 24.42 | 2.42 | 3.35 | 0.17 | 119 |
| IMERG-F V07 | 0.29 | 0.50 | 1.16 | 0.27 | 0.76 | 0.24 | 0.73 | -8.96 | 28.78 | 10.92 | 10.92 | 0.18 | 119 |
| model_16 | 0.28 | 0.40 | 1.04 | 0.32 | 0.95 | 0.17 | 0.66 | -15.42 | 38.45 | 1.53 | 3.47 | 0.14 | 119 |
| GDAS | 0.24 | 0.42 | 1.09 | 0.32 | 0.72 | 0.28 | 0.66 | -20.57 | 35.23 | 12.80 | 12.80 | 0.12 | 119 |
| model_17 | 0.23 | 0.38 | 0.89 | 0.37 | 0.94 | 0.18 | 0.62 | -9.37 | 40.42 | 0.44 | 3.75 | 0.11 | 119 |
| CMORPH-RT | 0.21 | 0.39 | 0.69 | 0.40 | 0.97 | 0.21 | 0.56 | -44.55 | 48.38 | 1.66 | 4.18 | 0.10 | 119 |
| model_04 | 0.21 | 0.38 | 0.75 | 0.35 | 1.05 | 0.18 | 0.62 | -37.17 | 44.83 | 0.20 | 2.81 | 0.12 | 118 |
| ERA5 | 0.21 | 0.36 | 0.99 | 0.24 | 0.75 | 0.28 | 0.61 | -30.18 | 39.06 | 11.82 | 11.82 | 0.11 | 119 |
| MSWEP V2.8 | 0.20 | 0.42 | 1.03 | 0.31 | 0.64 | 0.38 | 0.67 | -28.65 | 44.58 | 18.77 | 18.77 | 0.11 | 118 |
| SM2RAIN-GPM | 0.18 | 0.40 | 0.79 | 0.44 | 0.66 | 0.36 | 0.60 | -47.16 | 53.43 | 10.75 | 11.09 | 0.00 | 84 |
| CHIRPS V2 | 0.17 | 0.30 | 0.96 | 0.28 | 0.72 | 0.28 | 0.55 | -27.82 | 38.12 | 6.25 | 6.42 | 0.09 | 118 |
| model_18 | 0.13 | 0.29 | 0.90 | 0.37 | 0.89 | 0.22 | 0.45 | -24.39 | 33.54 | 4.26 | 6.98 | 0.09 | 119 |
| JRA-3Q | 0.12 | 0.29 | 1.16 | 0.41 | 0.72 | 0.29 | 0.59 | -18.02 | 38.92 | 9.79 | 9.84 | 0.10 | 119 |
| PDIR-Now | 0.11 | 0.32 | 1.34 | 0.47 | 0.80 | 0.26 | 0.47 | -5.89 | 29.44 | 16.30 | 16.30 | 0.10 | 119 |
| model_05 | 0.03 | 0.31 | 0.55 | 0.53 | 1.15 | 0.25 | 0.51 | -60.18 | 64.33 | -2.55 | 3.85 | 0.09 | 119 |
| CPC-Unified | -0.03 | 0.21 | 0.53 | 0.54 | 1.13 | 0.23 | 0.51 | -47.80 | 51.57 | -1.84 | 4.06 | 0.04 | 113 |
| CHIRP V2 | -0.07 | 0.23 | 1.10 | 0.27 | 0.35 | 0.65 | 0.39 | -56.68 | 57.45 | 44.36 | 44.36 | 0.00 | 119 |
| GSMaP-MVK V8 | -0.13 | 0.46 | 1.98 | 0.98 | 0.81 | 0.23 | 0.62 | 44.70 | 45.54 | 15.74 | 15.74 | 0.19 | 119 |
| PERSIANN-CCS-CDR | -0.18 | 0.31 | 1.84 | 0.84 | 0.78 | 0.23 | 0.49 | 62.49 | 62.83 | 11.36 | 11.65 | 0.11 | 118 |
| CMORPH-RAW | -0.19 | 0.44 | 2.06 | 1.06 | 0.82 | 0.21 | 0.57 | 77.54 | 77.54 | 10.80 | 10.95 | 0.18 | 119 |
| SM2RAIN-ASCAT | -0.20 | 0.19 | 1.27 | 0.34 | 0.27 | 0.73 | 0.31 | -67.16 | 67.41 | 49.68 | 49.68 | 0.00 | 117 |
| SM2RAIN-CCI | -0.24 | 0.18 | 0.42 | 0.71 | 0.33 | 0.70 | 0.38 | -83.05 | 85.41 | 5.25 | 20.16 | 0.00 | 65 |
| IMERG-L V07 | -0.39 | 0.45 | 2.22 | 1.22 | 0.75 | 0.26 | 0.61 | 72.91 | 72.91 | 14.75 | 14.75 | 0.16 | 119 |
| IMERG-E V07 | -0.44 | 0.42 | 2.26 | 1.26 | 0.71 | 0.31 | 0.59 | 67.20 | 67.20 | 19.79 | 19.79 | 0.16 | 119 |
| PERSIANN-CCS | -0.72 | 0.31 | 2.54 | 1.54 | 0.67 | 0.33 | 0.39 | 114.09 | 114.09 | 15.57 | 15.57 | 0.09 | 119 |



**Table 5.** Spearman rank correlation coefficients between rain gauge attributes (related to location, climate, and topography; see Table 2) and performance scores (KGE, $r_{\mathrm{dly}}$, $\beta$, etc.; see Section 2.6) for six key $P$ products and ML model stacks. The correlations were computed using independent evaluation gauges in Saudi Arabia ($n = 119$).

| Model | Predictor | KGE | $r_{\mathrm{dly}}$ | $\beta$ | $|\beta-1|$ | $\gamma$ | $|\gamma-1|$ | $r_{\mathrm{mon}}$ | $B_{\mathrm{peak}}$ | $|B_{\mathrm{peak}}|$ | $B_{\mathrm{wet\,days}}$ | $|B_{\mathrm{wet\,days}}|$ | $\mathrm{CSI}_{10\,\mathrm{mm}}$ |
|---|---|---|---|---|---|---|---|---|---|---|---|---|---|
| | Lat | 0.09 | 0.39 | 0.26 | 0.03 | 0.04 | 0.17 | 0.17 | 0.31 | 0.09 | −0.30 | −0.38 | 0.40 |
| | Lon | 0.05 | 0.17 | −0.01 | 0.08 | 0.06 | −0.04 | 0.32 | 0.07 | 0.01 | 0.05 | 0.42 | −0.04 |
| model_01 | Pmean | −0.13 | −0.42 | −0.32 | 0.00 | 0.00 | −0.08 | −0.24 | −0.29 | −0.02 | 0.22 | 0.48 | −0.39 |
| | AI | 0.14 | 0.43 | 0.28 | −0.01 | 0.00 | 0.09 | 0.24 | 0.28 | 0.03 | −0.24 | −0.47 | 0.40 |
| | ETH | −0.13 | −0.42 | −0.13 | −0.08 | −0.13 | 0.03 | −0.33 | −0.21 | −0.07 | 0.34 | 0.36 | −0.28 |
| | Lat | 0.13 | 0.24 | 0.03 | −0.03 | 0.45 | −0.01 | 0.07 | 0.15 | 0.09 | −0.44 | −0.39 | 0.04 |
| | Lon | −0.06 | 0.06 | 0.22 | 0.20 | −0.10 | −0.05 | 0.12 | 0.19 | 0.12 | 0.03 | 0.34 | 0.02 |
| model_06 | Pmean | −0.05 | −0.20 | 0.01 | −0.02 | −0.36 | −0.04 | −0.13 | −0.12 | −0.13 | 0.32 | 0.48 | 0.03 |
| | AI | 0.08 | 0.24 | 0.00 | 0.01 | 0.40 | 0.02 | 0.12 | 0.15 | 0.12 | −0.35 | −0.48 | −0.01 |
| | ETH | 0.08 | −0.16 | −0.12 | −0.22 | −0.17 | 0.01 | −0.19 | −0.29 | −0.33 | 0.19 | 0.31 | 0.08 |
| | Lat | 0.47 | 0.68 | 0.34 | −0.05 | 0.19 | −0.24 | 0.53 | 0.50 | −0.21 | −0.71 | −0.72 | 0.67 |
| | Lon | −0.22 | −0.01 | −0.12 | 0.20 | 0.00 | 0.08 | 0.08 | −0.11 | 0.09 | 0.26 | 0.26 | −0.23 |
| ERA5 | Pmean | −0.44 | −0.64 | −0.33 | 0.11 | 0.00 | 0.01 | −0.48 | −0.42 | 0.18 | 0.69 | 0.71 | −0.57 |
| | AI | 0.47 | 0.67 | 0.32 | −0.08 | 0.06 | −0.06 | 0.51 | 0.42 | −0.20 | −0.72 | −0.73 | 0.58 |
| | ETH | −0.22 | −0.50 | −0.15 | 0.01 | −0.02 | −0.02 | −0.41 | −0.28 | 0.01 | 0.51 | 0.53 | −0.36 |
| | Lat | 0.25 | 0.40 | 0.12 | −0.05 | 0.43 | −0.37 | 0.24 | 0.47 | −0.12 | −0.81 | −0.81 | 0.19 |
| | Lon | −0.08 | 0.05 | −0.20 | 0.14 | 0.15 | 0.05 | 0.13 | −0.23 | 0.12 | 0.16 | 0.18 | −0.22 |
| MSWEP V2.8 | Pmean | −0.17 | −0.29 | −0.21 | −0.02 | −0.35 | 0.29 | −0.17 | −0.60 | 0.17 | 0.79 | 0.80 | −0.18 |
| | AI | 0.21 | 0.29 | 0.16 | −0.01 | 0.37 | −0.29 | 0.17 | 0.57 | −0.15 | −0.82 | −0.83 | 0.19 |
| | ETH | −0.03 | −0.22 | −0.12 | −0.19 | −0.36 | 0.21 | −0.20 | −0.48 | 0.12 | 0.60 | 0.60 | −0.01 |
| | Lat | −0.41 | 0.24 | 0.42 | 0.42 | −0.14 | 0.16 | −0.08 | 0.48 | 0.48 | −0.31 | −0.31 | −0.03 |
| | Lon | 0.05 | 0.18 | −0.02 | −0.02 | 0.02 | 0.04 | 0.40 | −0.11 | −0.13 | 0.43 | 0.44 | 0.08 |
| IMERG−L V07 | Pmean | 0.52 | −0.31 | −0.52 | −0.52 | 0.27 | −0.29 | 0.04 | −0.61 | −0.61 | 0.28 | 0.28 | 0.08 |
| | AI | −0.53 | 0.29 | 0.53 | 0.54 | −0.25 | 0.27 | −0.07 | 0.62 | 0.62 | −0.27 | −0.27 | −0.08 |
| | ETH | 0.39 | −0.37 | −0.41 | −0.40 | 0.33 | −0.37 | −0.15 | −0.47 | −0.46 | 0.08 | 0.08 | 0.09 |
| | Lat | −0.21 | 0.39 | 0.23 | 0.23 | −0.27 | 0.12 | 0.20 | 0.20 | 0.19 | −0.48 | −0.48 | 0.21 |
| | Lon | 0.32 | 0.14 | −0.27 | −0.26 | 0.27 | −0.20 | 0.38 | −0.19 | −0.15 | 0.08 | 0.08 | 0.25 |
| GSMaP-MVK V8 | Pmean | 0.36 | −0.33 | −0.38 | −0.37 | 0.40 | −0.23 | −0.20 | −0.32 | −0.30 | 0.48 | 0.48 | −0.05 |
| | AI | −0.33 | 0.35 | 0.35 | 0.35 | −0.37 | 0.21 | 0.20 | 0.32 | 0.30 | −0.49 | −0.49 | 0.08 |
| | ETH | 0.07 | −0.34 | −0.12 | −0.12 | 0.16 | −0.03 | −0.35 | −0.17 | −0.15 | 0.54 | 0.54 | −0.06 |





**Table A1.** Hyperparameters for XGBoost model.

| Hyperparameter | Value |
| --- | --- |
| n_estimators | 100 |
| max_depth | 12 |
| min_child_weight | 5 |
| colsample_bytree | 0.7 |
| gamma | 2 |
| reg_alpha | 0.5 |
| reg_lambda | 0.5 |
| learning_rate | 0.2 |

**Table A2.** Hyperparameters for RF model.

| Hyperparameter | Value |
| --- | --- |
| n_estimators | 100 |
| max_depth | 15 |
| min_samples_split | 5 |
| min_samples_leaf | 5 |
| max_features | 0.7 |