# Peer review of "Saudi Rainfall (SaRa): Hourly 0.1° Gridded Rainfall (1979–Present) for Saudi Arabia via Machine Learning Fusion of Satellite and Model Data"

_EGUsphere, 2025_

## Author Comment (AC1)

This study presents a machine-learning-based approach to estimating gridded rainfall data for Saudi Arabia, an arid region with significant data limitations. The proposed dataset, SaRa, is compared against multiple existing precipitation datasets. The approach uses a combination of random forests and XGboots models. While not very novel, the results suggest superior performance, thus adding value and contributing to the data availability in the region. While the paper is well-structured with a sound methodology, fundamental concerns arise regarding the model accuracy away from training sites, generalizability, and reliability of the identified trends.

We thank the reviewer for their insightful comments.

Major:

1. I appreciate the authors filtering for potentially double precipitation gauges within 2 km, but the paper needs more clarity on how the split training/testing sample was performed. Was it random? Stratified? Distance-based?

Good question. This was done randomly. Since we had so many stations, we decided to allocate a very large amount (50%) to the validation subset, to ensure a robust validation. We have explicitly mentioned this in the text.

2. When applying ML to geospatial datasets, a critical issue is the use of testing sites near training sites that often artificially boost validation statistics. That's because precipitation data is spatially correlated. To enhance transparency and thrust into ML approaches, the accuracy of the ML models should also be evaluated based on their distance from training sites. Please plot the KGE testing accuracy of each testing point vs. its distance from the nearest training site (km). This will evidence how well the proposed ML approach is trusted in distant/ungauged areas. This plot would be informative for the main individual ML models and the ensemble stack.

Thanks for the great suggestions. We will include a new figure in the paper showing KGE values of testing stations versus distance to the nearest training station. This plot indicates that the KGE value does not decrease with distance to the nearest training station, underscoring the generalizability of our approach. We will also add corresponding text to section 3.1:

"A key limitation of ML-based P estimation is poor generalizability; models often fail in regions lacking training data (Xu et al., 2024). To assess whether this applies to our models, we analyzed KGE values of the evaluation stations as a function of distance to the nearest training station (Figure 4). The results show no clear decline in KGE with increasing distance, indicating satisfactory spatial generalizability."

3. The ensemble approach, while interesting, results in a black-box system—there is little discussion on the physical interpretability of the model structures and the predictive power of the inputs. Sklearn Random forests and XGboost have out-of-the-box libraries that can be easily deployed to evaluate model interpretability further. This could improve model understanding and expand the proposed approaches' generalizability.

We have calculated the importance of each predictor for each of the 18 model stacks and each of the four submodels. We cannot present all importances for all model stacks and submodels, as this would overwhelm the reader. However, we agree that physical interpretability of the model is important; therefore, we will add a new table to present importances for model_01 for all of its four submodels (Table 5). We will also add corresponding text to section 3.1 to discuss the results:

*"To improve transparency, we computed predictor importances for all four submodels of model_01 (Table 5). IMERG-L V07 consistently ranked higher than GSMaP-MVK V8 in importance, indicating a model preference for IMERG, which aligns with its superior validation performance (Table 4). ERA5 was the most important predictor for the daily submodel, whereas IMERG dominated in the 3-hourly and hourly submodels. This likely reflects the advantage of observational datasets like IMERG in accurately capturing event timing. Static predictors were overall much less important than dynamic ones. Among the static predictors, abs_lat and lat had the highest importances, likely reflecting the latitudinal dependence of P product performance observed in global evaluations (e.g., Beck et al., 2017)."*

4. The study does not sufficiently address uncertainty in trend estimations. There are no confidence intervals, no discussion of interannual variability, and no attempt to separate natural variability from long-term trends. Given the known issues with historical precipitation datasets, particularly in arid regions, one must question how much of the trend results from dataset evolution rather than actual climate change.

Thank you for the comment. We already strongly emphasize the uncertainty in the trends and also highlight the interannual variability in the trend section of the paper (section 3.5), however, to emphasize this further, we will add the following text:

*"However, it is important to note that these trend estimates are subject to significant uncertainty due to considerable interannual variability, as well as substantial uncertainties in gauge, model, and satellite P estimates (see Section 3.4). In addition to random errors, satellite datasets are affected by transitions in data sources and radar sensors used for calibration (e.g., TRMM to GPM circa 2015; see Huffman, 2019), while reanalyses are affected by updates in data assimilation, such as the progressive inclusion of new satellite datasets (e.g., the TOVS to ATOVS transition in 2000), as well as the concatenation of*

*different production streams (Hersbach, 2020). These discontinuities propagate through and are reflected in SaRa. Interestingly, future projections from climate models in the sixth phase of the Coupled Model Intercomparison Project (CMIP6) indicate that increases in all three metrics are likely across most regions of Saudi Arabia (Iturbide et al., 2021; Calvin et al., 2023)"*

Furthermore, to better convey the uncertainty in the trends, we will add dots to the map (Figure 6) to indicate grid-cells with significant trends (p<0.05).

Unfortunately, it is not straightforward to separate natural variability from long-term trends. Since long-term trends can also be natural, we assume the reviewer means anthropogenic impacts. However, these are extremely difficult to quantify for precipitation, requiring long observational records, convection-permitting climate simulations, and advanced attribution analysis.

Regarding the comment on separating "dataset evolution" from actual climate change, as mentioned in the Data and Methods section, time series from different model stacks are harmonized to our reference model stack, model_01. As such, the time series should be relatively homogenous through time.

Moderate:

1. The paper would benefit from a quantitative analysis and discussion of how temporal resolution mismatches in the gauge data impact validation results.

We do not think temporal resolution mismatches exert a major impact on validation results, because all the gridded datasets were either in daily temporal resolution originally, or have been aggregated to the daily resolution.

If the reviewer is referring to mismatches in gauge reporting times, it is true that reporting times may impact the validation, namely degrade the results for gauges with large mismatches. However, this phenomenon affects all datasets equally; hence it does not affect the performance ranking of our datasets. Note that we have highlighted the issue of reporting times in section 3.4:

*"Time shifts between daily P totals from gauges and satellite or (re)analysis products (Yang et al., 2020; Beck et al., 2019) further reduce performance scores, especially in arid regions due to the short duration of rainfall events. The boundary between daily totals from satellite or (re)analysis products is midnight UTC, whereas it varies for daily gauge totals depending on regional reporting practices. In Saudi Arabia, the average boundary time was determined to be 05:00 AM UTC (08:00 AM local time; see Section 2.6). Consequently, for a short, hourly event, there is a $100 \times 5/24 = 21$ % chance that it will be assigned to the 'wrong' day."*

Minor:

1. L 143 clarify what are gross errors.

Gross errors are unpredictable mistakes from careless observers when using equipment, reading scales or recording observations (Thapa and Bossler, 1992).

**References**

Beck, H. E., Vergopolan, N., Pan, M., Levizzani, V., Van Dijk, A. I., Weedon, G. P., ... & Wood, E. F. (2017). Global-scale evaluation of 22 precipitation datasets using gauge observations and hydrological modeling. Hydrology and Earth System Sciences, 21(12), 6201-6217.

Beck, H. E., Wood, E. F., Pan, M., Fisher, C. K., Miralles, D. G., Van Dijk, A. I., ... & Adler, R. F. (2019). MSWEP V2 global 3-hourly 0.1 precipitation: methodology and quantitative assessment. Bulletin of the American Meteorological Society, 100(3), 473-500.

Calvin, K., Dasgupta, D., Krinner, G., Mukherji, A., Thorne, P. W., Trisos, C., ... & Hauser, M. (2023). IPCC, 2023: Climate Change 2023: Synthesis Report, Summary for Policymakers. Contribution of Working Groups I, II and III to the Sixth Assessment Report of the Intergovernmental Panel on Climate Change [Core Writing Team, H. Lee and J. Romero (eds.)]. IPCC, Geneva, Switzerland. IPCC, 2023: Climate Change 2023: Synthesis Report. Contribution of Working Groups I, II and III to the Sixth Assessment Report of the Intergovernmental Panel on Climate Change [Core Writing Team, H. Lee and J. Romero (eds.)]. IPCC, Geneva, Switzerland., 1-34.

Hersbach, H., Bell, B., Berrisford, P., Hirahara, S., Horányi, A., Muñoz-Sabater, J., ... & Thépaut, J. N. (2020). The ERA5 global reanalysis. Quarterly journal of the royal meteorological society, 146(730), 1999-2049.

Huffman, G. J. (2019). The transition in multi-satellite products from TRMM to GPM (TMPA to IMERG). Algorithm Information Document.

Thapa, K., & Bossler, J. (1992). Accuracy of spatial data used information systems. Photogrammetric Engineering & Remote Sensing, 58(6), 835-841.

Xu, Y., Tang, G., Li, L., & Wan, W. (2024). Multi-source precipitation estimation using machine learning: Clarification and benchmarking. Journal of Hydrology, 635, 131195.

Yang, S., Jones, P. D., Jiang, H., & Zhou, Z. (2020). Development of a near-real-time global in situ daily precipitation dataset for 0000–0000 UTC. International Journal of Climatology, 40(5), 2795-2810.

---

## Author Comment (AC2)

**Response to RC2**

Review for " Saudi Rainfall (SaRa): Hourly 0.1° Gridded Rainfall (1979–Present) for Saudi Arabia via Machine Learning Fusion of Satellite and Model Data" by Wang et al. submitted to EGUsphere (MS No.: egusphere-2025-254).

General comments:

The authors introduce *Saudi Rainfall (SaRa)*, a gridded precipitation product for the Arabian Peninsula developed using Machine Learning (ML) techniques. They clearly present the motivation behind the development of such a dataset, describe the procedures used to generate the SaRa product, and evaluate its performance. By leveraging a large amount of available gauge-based and gridded datasets, the authors produce a new dataset that shows improved performance compared to existing products—particularly in areas with sparse station observations and in the dominantly arid regions of the Arabian Peninsula.

This work makes a valuable contribution to the data community and enhances scientific understanding of precipitation patterns in data-scarce, arid environments. The overall quality of the manuscript is good, with well-cited references and generally clear writing.

We thank the reviewer for their useful feedback.

However, there is still room for further improvement. In particular, I would like to raise two main concerns:

1. Limitations of Machine Learning: What are the potential limitations, challenges and sources of error introduced by using Machine Learning techniques in generating this dataset? A discussion on uncertainties and biases associated with ML itself would strengthen the paper.

This is a great comment. We agree that ML is subject to some limitations, but as our independent validation shows, our ML model outperforms all other gridded datasets for all metrics. Nevertheless, to address this comment, we will add the following text to section 3.1:

*"A key limitation of ML-based P estimation is poor generalizability; models often fail in regions lacking training data (Xu et al., 2024). To assess whether this applies to our models, we analyzed KGE values of the evaluation stations as a function of distance to the nearest training station (Figure 4). The results show no clear decline in KGE with increasing distance, indicating satisfactory spatial generalizability. Another common criticism is the "black-box" nature of ML models, which limits interpretability.*

*To improve transparency, we computed predictor importances for all four submodels of model_01 (Table 5). IMERG-L V07 consistently ranked higher than GSMaP-MVK V8 in importance, indicating a model preference for IMERG, which aligns with its superior validation performance (Table 4). ERA5 was the most important predictor for the daily submodel, whereas IMERG dominated in the 3-hourly and hourly submodels. This likely reflects the advantage of observational datasets like IMERG in accurately capturing event timing. Static predictors were overall much less important than dynamic ones. Among the static predictors, abs_lat and lat had the highest importances, likely reflecting the latitudinal dependence of P product performance observed in global evaluations (e.g., Beck et al., 2017)."*

A third challenge is that ML methods inherently tend to underestimate extreme precipitation events due to regression toward the mean. This is discussed in sections 2.5 and 3.1, and was addressed in SaRa by the inclusion of submodel 3, which corrects the CDF.

2. Broader Impact and Global Appeal: What is the relevance of this work beyond the Arabian Peninsula? Discussing the broader applicability of the methodology and insights would enhance the global significance of the study.

This is a great point! We believe the algorithm we used in this paper could also be applied globally. We are actually also working on the new version of MSWEP (Multi-Source Weighted-Ensemble Precipitation) by using the SaRa algorithm as the base. To highlight the broader applicability, we have the following text:

*"The SaRa dataset, available for use and distribution at www.gloh2o.org/sara, equips researchers, professionals, and policymakers with the tools needed to tackle pressing environmental and socio-economic challenges in Saudi Arabia, and serves as a potential framework for filling this data gap in other arid and dryland regions."*

We will not mention MSWEP in the current context as it is still in development.

In addition, I suggest the authors consider the following points:

Include a study area map: Add a map of the Arabian Peninsula showing the region's topography and its location in a global context. This would help orient readers unfamiliar with the area.

We will include a new figure to indicate the study area

Describe ML Challenges: Provide a more detailed discussion of the challenges and limitations in implementing ML for P data generation.

We will add a detailed paragraph discussing the main limitations of ML for precipitation estimation. Please see our response to the reviewer's first point.

Discuss Practical Applications: Expand the discussion to highlight potential applications of the dataset, such as its use in flash flood risk mitigation, water resource management, or climate-related decision-making in arid regions.

Thank you. We have the following text in the paper which we believe addresses your comment:

*"The SaRa dataset, available for use and distribution at www.gloh2o.org/sara, equips researchers, professionals, and policymakers with the tools needed to tackle pressing environmental and socio-economic challenges in Saudi Arabia, and serves as a potential framework for filling this data gap in other arid and dryland regions. The product delivers a high-resolution, near real-time resource designed to support a diverse range of applications, including water resource management, hydrological modeling, agricultural planning, disaster risk reduction, and climate studies."*

References

Beck, H. E., Vergopolan, N., Pan, M., Levizzani, V., Van Dijk, A. I., Weedon, G. P., ... & Wood, E. F. (2017). Global-scale evaluation of 22 precipitation datasets using gauge observations and hydrological modeling. Hydrology and Earth System Sciences, 21(12), 6201-6217.

Xu, Y., Tang, G., Li, L., & Wan, W. (2024). Multi-source precipitation estimation using machine learning: Clarification and benchmarking. Journal of Hydrology, 635, 131195.

---

## Author Response (AR1)

**Response to RC1**

This study presents a machine-learning-based approach to estimating gridded rainfall data for Saudi Arabia, an arid region with significant data limitations. The proposed dataset, SaRa, is compared against multiple existing precipitation datasets. The approach uses a combination of random forests and XGboots models. While not very novel, the results suggest superior performance, thus adding value and contributing to the data availability in the region. While the paper is well-structured with a sound methodology, fundamental concerns arise regarding the model accuracy away from training sites, generalizability, and reliability of the identified trends.

We thank the reviewer for their insightful comments.

Major:

1. I appreciate the authors filtering for potentially double precipitation gauges within 2 km, but the paper needs more clarity on how the split training/testing sample was performed. Was it random? Stratified? Distance-based?

Good question. This was done randomly. Since we had so many stations, we decided to allocate a very large amount (50%) to the validation subset, to ensure a robust validation. We have explicitly mentioned this in the text. This explanation is now explicitly mentioned in L184.

2. When applying ML to geospatial datasets, a critical issue is the use of testing sites near training sites that often artificially boost validation statistics. That's because precipitation data is spatially correlated. To enhance transparency and thrust into ML approaches, the accuracy of the ML models should also be evaluated based on their distance from training sites. Please plot the KGE testing accuracy of each testing point vs. its distance from the nearest training site (km). This will evidence how well the proposed ML approach is trusted in distant/ungauged areas. This plot would be informative for the main individual ML models and the ensemble stack.

Thanks for the great suggestions. We have included a new figure (Figure 5) in the paper showing KGE values of testing stations versus distance to the nearest training station. This plot indicates that the KGE value does not decrease with distance to the nearest training station, underscoring the generalizability of our approach. We have also added corresponding text to section 3.1:

*"A key potential limitation of ML-based P estimation is poor generalizability; models often fail in regions lacking training data (Xu et al., 2024). To assess whether this applies to our models, we analyzed KGE values of the evaluation stations as a function of distance to the*

*nearest training station (Figure 5). The results show no clear decline in KGE with increasing distance, indicating satisfactory spatial generalizability.”*

3. The ensemble approach, while interesting, results in a black-box system—there is little discussion on the physical interpretability of the model structures and the predictive power of the inputs. Sklearn Random forests and XGboost have out-of-the-box libraries that can be easily deployed to evaluate model interpretability further. This could improve model understanding and expand the proposed approaches' generalizability.

We have calculated the importance of each predictor for each of the 18 model stacks and each of the four submodels. We cannot present all importances for all model stacks and submodels, as this would overwhelm the reader. However, we agree that physical interpretability of the model is important; therefore, we have added a new table to present importances for model_01 for all of its four submodels (Table 6). We have also added corresponding text to section 3.1 to discuss the results:

*“To improve transparency, we computed predictor importances for all four submodels of model_01 (Table 6). IMERG-L V07 consistently ranked higher than GSMaP-MVK V8 in importance, indicating a preference for IMERG, in agreement with its superior validation performance (Table 4). ERA5 was the most important predictor for the daily submodel (Submodel 1), whereas IMERG dominated in the 3-hourly and hourly submodels (Submodels 2 and 4, respectively). This likely reflects the ability of observational satellite-based datasets like IMERG to capture event timing more accurately. This also demonstrates the ability of the models to exploit the complementary strengths of the different P predictors. Static predictors were overall much less important than dynamic ones. Among the static predictors, Lon, Lat, and AbsLat had the highest importances, accounting for regional variability in P predictor performance and error characteristics.”*

4. The study does not sufficiently address uncertainty in trend estimations. There are no confidence intervals, no discussion of interannual variability, and no attempt to separate natural variability from long-term trends. Given the known issues with historical precipitation datasets, particularly in arid regions, one must question how much of the trend results from dataset evolution rather than actual climate change.

Thank you for the comment. We already strongly emphasize the uncertainty in the trends and also highlight the interannual variability in the trend section of the paper (section 3.5), however, to emphasize this further, we have added the following text (L384–395):

*“However, it is important to note that these trend estimates are mostly statistically insignificant (p-value > 0.05) and subject to substantial uncertainty due to large interannual*

*variability, as well as considerable errors in gauge, reanalysis, and satellite P estimates (see Section 3.4). In addition to random errors, satellite datasets are affected by transitions in data sources and radar sensors used for calibration (e.g., TRMM to GPM circa 2015; see Huffman, 2019), while reanalyses are affected by updates in data assimilation, such as the progressive inclusion of new satellite datasets (e.g., the TOVS to ATOVS transition in 2000), as well as the concatenation of different production streams (Hersbach et al., 2020). These discontinuities propagate through and are reflected in SaRa, contributing to the observed uncertainties and hindering the detection of significant trends. Despite historical declines, future projections from climate models in the sixth phase of the Coupled Model Intercomparison Project (CMIP6) indicate that increases in all three metrics (mean annual P, annual maximum daily P, and annual rainy days) are likely across most regions of Saudi Arabia (Iturbide et al., 2021; Intergovernmental Panel on Climate Change (IPCC), 2023), highlighting the need and value of data-driven P approaches in resolving potential discrepancies in P distributions and spatio-temporal patterns."*

Furthermore, to better convey the uncertainty in the trends, we have added dots to the map (Figure 7) to indicate grid-cells with significant trends (p < 0.05).

Unfortunately, it is not straightforward to separate natural variability from long-term trends. Since long-term trends can also be natural, we assume the reviewer means anthropogenic impacts. However, these are extremely difficult to quantify for precipitation, requiring long observational records, convection-permitting climate simulations, and advanced attribution analysis.

Regarding the comment on separating "dataset evolution" from actual climate change, as mentioned in the Data and Methods section, time series from different model stacks are harmonized to our reference model stack, model_01. As such, the time series should be relatively homogenous through time.

Moderate:

1. The paper would benefit from a quantitative analysis and discussion of how temporal resolution mismatches in the gauge data impact validation results.

We do not think temporal resolution mismatches exert a major impact on validation results, because all the gridded datasets were either in daily temporal resolution originally, or have been aggregated to the daily resolution.

If the reviewer is referring to mismatches in gauge reporting times, it is true that reporting times may impact the validation, namely degrade the results for gauges with large mismatches. However, this phenomenon affects all datasets equally; hence it does not affect the performance ranking of our datasets. Note that we have highlighted the issue of reporting times in section 3.4:

*"Time shifts between daily P totals from gauges and satellite or (re)analysis products (Yang et al., 2020; Beck et al., 2019) further reduce performance scores, especially in arid regions due to the short duration of rainfall events. The boundary between daily totals from satellite or (re)analysis products is midnight UTC, whereas it varies for daily gauge totals depending on regional reporting practices. In Saudi Arabia, the average boundary time was determined to be 05:00 AM UTC (08:00 AM local time; see Section 2.6). Consequently, for a brief event of one hour, there is a 100 × 5/24 = 21 % chance that it will be assigned to the 'wrong' day."*

Minor:

1. L 143 clarify what are gross errors.

Gross errors are unpredictable mistakes from careless observers when using equipment, reading scales or recording observations (Thapa and Bossler, 1992). But thanks for the suggestion and we have replaced "gross" by "human-made" in the revised manuscript.

**References**

Beck, H. E., Wood, E. F., Pan, M., Fisher, C. K., Miralles, D. G., Van Dijk, A. I., ... & Adler, R. F. (2019). MSWEP V2 global 3-hourly 0.1 precipitation: methodology and quantitative assessment. Bulletin of the American Meteorological Society, 100(3), 473-500.

Calvin, K., Dasgupta, D., Krinner, G., Mukherji, A., Thorne, P. W., Trisos, C., ... & Hauser, M. (2023). IPCC, 2023: Climate Change 2023: Synthesis Report, Summary for Policymakers. Contribution of Working Groups I, II and III to the Sixth Assessment Report of the Intergovernmental Panel on Climate Change [Core Writing Team, H. Lee and J. Romero (eds.)]. IPCC, Geneva, Switzerland. IPCC, 2023: Climate Change 2023: Synthesis Report. Contribution of Working Groups I, II and III to the Sixth Assessment Report of the Intergovernmental Panel on Climate Change [Core Writing Team, H. Lee and J. Romero (eds.)]. IPCC, Geneva, Switzerland., 1-34.

Hersbach, H., Bell, B., Berrisford, P., Hirahara, S., Horányi, A., Muñoz-Sabater, J., ... & Thépaut, J. N. (2020). The ERA5 global reanalysis. Quarterly journal of the royal meteorological society, 146(730), 1999-2049.

Huffman, G. J. (2019). The transition in multi-satellite products from TRMM to GPM (TMPA to IMERG). Algorithm Information Document.

Thapa, K., & Bossler, J. (1992). Accuracy of spatial data used information systems. Photogrammetric Engineering & Remote Sensing, 58(6), 835-841.

Xu, Y., Tang, G., Li, L., & Wan, W. (2024). Multi-source precipitation estimation using machine learning: Clarification and benchmarking. Journal of Hydrology, 635, 131195.

Yang, S., Jones, P. D., Jiang, H., & Zhou, Z. (2020). Development of a near-real-time global in situ daily precipitation dataset for 0000–0000 UTC. International Journal of Climatology, 40(5), 2795-2810.

**Response to RC2**

Review for " Saudi Rainfall (SaRa): Hourly 0.1° Gridded Rainfall (1979–Present) for Saudi Arabia via Machine Learning Fusion of Satellite and Model Data" by Wang et al. submitted to EGUsphere (MS No.: egusphere-2025-254).

General comments:

The authors introduce *Saudi Rainfall (SaRa)*, a gridded precipitation product for the Arabian Peninsula developed using Machine Learning (ML) techniques. They clearly present the motivation behind the development of such a dataset, describe the procedures used to generate the SaRa product, and evaluate its performance. By leveraging a large amount of available gauge-based and gridded datasets, the authors produce a new dataset that shows improved performance compared to existing products—particularly in areas with sparse station observations and in the dominantly arid regions of the Arabian Peninsula.

This work makes a valuable contribution to the data community and enhances scientific understanding of precipitation patterns in data-scarce, arid environments. The overall quality of the manuscript is good, with well-cited references and generally clear writing.

We thank the reviewer for their useful feedback.

However, there is still room for further improvement. In particular, I would like to raise two main concerns:

1. Limitations of Machine Learning: What are the potential limitations, challenges and sources of error introduced by using Machine Learning techniques in generating this dataset? A discussion on uncertainties and biases associated with ML itself would strengthen the paper.

This is a great comment. We agree that ML is subject to some limitations, but as our independent validation shows, our ML model outperforms all other gridded datasets for all metrics. Nevertheless, to address this comment, in combination with those of Reviewer 1, we have added the following text to section 3.1:

*"A key potential limitation of ML-based P estimation is poor generalizability; models often fail in regions lacking training data (Xu et al., 2024). To assess whether this applies to our models, we analyzed KGE values of the evaluation stations as a function of distance to the nearest training station (Figure 5). The results show no clear decline in KGE with increasing distance, indicating satisfactory spatial generalizability. Another potential limitation is the "black-box" nature of ML models, which limits interpretability. To improve transparency, we computed predictor importances for all four submodels of model_01 (Table 6). IMERG-L V07 consistently ranked higher than GSMaP-MVK V8 in importance, indicating a preference for IMERG, in agreement with its superior validation performance (Table 4). ERA5 was the most important predictor for the daily submodel (Submodel 1), whereas IMERG dominated in the 3-hourly and hourly submodels (Submodels 2 and 4, respectively). This likely reflects the ability of observational satellite-based datasets like IMERG to capture event timing more accurately. This also demonstrates the ability of the models to exploit the complementary strengths of the different P predictors. Static predictors were overall much less important than dynamic ones. Among the static predictors, Lon, Lat, and AbsLat had the highest importances, accounting for regional variability in P predictor performance and error characteristics."*

Another challenge is that ML methods inherently tend to underestimate extreme precipitation events due to regression toward the mean. This is discussed in sections 2.5 and 3.1, and was addressed in SaRa by the inclusion of submodel 3, which corrects the CDF.

2. Broader Impact and Global Appeal: What is the relevance of this work beyond the Arabian Peninsula? Discussing the broader applicability of the methodology and insights would enhance the global significance of the study.

This is a great point! We believe the algorithm we used in this paper could also be applied globally. We are actually also working on the new version of MSWEP (Multi-Source Weighted-Ensemble Precipitation) by using the SaRa algorithm as the base. To highlight the broader applicability, we have the following text:

*"The SaRa dataset, available at www.gloh2o.org/sara, equips researchers, professionals, and policymakers with the tools needed to tackle pressing environmental and socio-economic challenges in Saudi Arabia, and serves not only as a potential framework for filling this data gap in other arid and dryland regions, but as a framework that could be applied globally to develop a consistent long-term dataset."*

We will not mention MSWEP in the current context as it is still in development.

In addition, I suggest the authors consider the following points:

Include a study area map: Add a map of the Arabian Peninsula showing the region's topography and its location in a global context. This would help orient readers unfamiliar with the area.

We have included a new figure (Figure 1) to indicate the study area

Describe ML Challenges: Provide a more detailed discussion of the challenges and limitations in implementing ML for P data generation.

We have added a detailed paragraph discussing the main limitations of ML for precipitation estimation. Please see our response to the reviewer's first point.

Discuss Practical Applications: Expand the discussion to highlight potential applications of the dataset, such as its use in flash flood risk mitigation, water resource management, or climate-related decision-making in arid regions.

Thank you. We have the following text in the paper which we believe addresses your comment:

*"The SaRa dataset, available at www.gloh2o.org/sara, equips researchers, professionals, and policymakers with the tools needed to tackle pressing environmental and socio-economic challenges in Saudi Arabia, and serves not only as a potential framework for filling this data gap in other arid and dryland regions, but as a framework that could be applied globally to develop a consistent long-term dataset. The product delivers a high-resolution, near real-time resource designed to support a diverse range of applications, including water resource management, hydrological modeling, agricultural planning, disaster risk reduction, and climate studies."*

References

Xu, Y., Tang, G., Li, L., & Wan, W. (2024). Multi-source precipitation estimation using machine learning: Clarification and benchmarking. Journal of Hydrology, 635, 131195.